# Trajectory-Class-Aware Multi-Agent Reinforcement Learning

**Hyungho Na**[1]**, Kwanghyeon Lee**[1]**, Sumin Lee** [1]**, Il-Chul Moon**[1,2]
[1]Korea Advanced Institute of Science and Technology (KAIST), [2]summary.ai
{gudgh723}@gmail.com,{rhkdgus0414,sumlee,icmoon}@kaist.ac.kr

## Abstract

In the context of multi-agent reinforcement learning, *generalization* is a challenge to solve various tasks that may require different joint policies or coordination without relying on policies specialized for each task. We refer to this type of problem as a *multi-task*, and we train agents to be versatile in this multi-task setting through a single training process. To address this challenge, we introduce **TR**ajectory-class-**A**ware **M**ulti-**A**gent reinforcement learning (**TRAMA**). In TRAMA, agents recognize a task type by identifying the class of trajectories they are experiencing through partial observations, and the agents use this trajectory awareness or prediction as additional information for action policy. To this end, we introduce three primary objectives in TRAMA: (a) constructing a quantized latent space to generate trajectory embeddings that reflect key similarities among them; (b) conducting trajectory clustering using these trajectory embeddings; and (c) building a trajectory-class-aware policy. Specifically for (c), we introduce a trajectory-class predictor that performs agent-wise predictions on the trajectory class; and we design a trajectory-class representation model for each trajectory class. Each agent takes actions based on this trajectory-class representation along with its partial observation for task-aware execution. The proposed method is evaluated on various tasks, including multi-task problems built upon StarCraft II. Empirical results show further performance improvements over state-of-the-art baselines.

## 1 Introduction

The value factorization framework (Sunehag et al., 2017; Rashid et al., 2018; Wang et al., 2020a) under Centralized Training with Decentralized Execution (CTDE) paradigm (Oliehoek et al., 2008; Gupta et al., 2017) has demonstrated its effectiveness across a range of cooperative multi-agent tasks (Lowe et al., 2017; Samvelyan et al., 2019). However, learning optimal policy often takes a long training time in more complex tasks, and the trained model often falls into suboptimal policies. These suboptimal results are often observed in complex tasks, which require agents to search large joint action-observation spaces.

Researchers have introduced task division methods in diverse frameworks to overcome this limitation. Although previous works have used different terminologies, such as skills (Yang et al., 2019; Liu et al., 2022), subtasks (Yang et al., 2022), and roles (Wang et al., 2020b; 2021); they have shared the major objective, such as reducing the search space of each agent during training or encouraging committed behavior for coordination among agents. For this purpose, agents first determine its role, skill, or subtask often by upper-tier policies (Wang et al., 2021; Liu et al., 2022; Yang et al., 2022); and the agents determine actions by this additional condition along with their partial observations. Compared to common MARL approaches, these task division methods show a strong performance in some complex tasks.

Recently, *multi-agent multi-tasks* have become new challenges in generalizing the learned policy to be effective in diverse settings. These tasks require agents to learn versatile policies for solving distinct problems, which may demand different joint policies or coordination among agents through a single MARL training process. For example, in SMACv2 (Ellis et al., 2024), agents need to learn policies neutralizing enemies in different initial positions and even with different unit combinations unlike the original SMAC (Samvelyan et al., 2019). In this new challenge, the previous task division

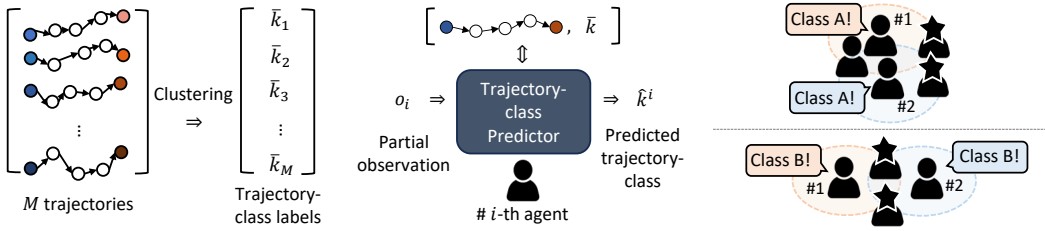

| (a) Trajectory Clustering | (b) Trajectory-Class Prediction | (c) Trajectory-Class-Aware Policy |

Figure 1: Illustration of the overall procedure for trajectory-class-aware policy learning: (a) through trajectory clustering, each trajectory is labeled. (b) Each agent predicts which trajectory class it is experiencing based on its partial observation. (c) After identifying the trajectory class, agents perform trajectory-class-dependent decision-making. In (c), each agent succeeds in identifying the same trajectory class based on its partial observations denoted with different colors.

approaches show unsatisfactory performance because a specialized policy for a single task will not work under different settings.

Motivated by this new challenge and the limitations of state-of-the-art MARL algorithms, we develop a framework that enables agents to recognize task types during task execution. The agents then use these task-type predictions or conditions in decision-making for versatile policy learning.

**Contribution.** This paper presents **TR**ajectory-class-**A**ware **M**ulti-**A**gent reinforcement learning (**TRAMA**) to perform the newly suggested functionality described below.

- **Constructing a quantized latent space for trajectory embedding:** To generate trajectory embeddings that reflect key similarities among them, we adopt the quantized latent space via Vector Quantized-Variational Autoencoder (VQ-VAE) (Van Den Oord et al., 2017). However, the naive adaptation of VQ-VAE for state embedding in MARL results in sparse usage of quantized vectors highlighted by (Na & Moon, 2024). To address this problem in multi-task settings, we introduce modified *coverage loss* considering trajectory class in VQ-VAE training to spread quantized vectors evenly throughout the embedding space of feasible states.

- **Trajectory clustering:** In TRAMA, we conduct trajectory clustering based on trajectory embeddings in the quantized latent space to identify trajectories that share key similarities. Since we cannot conduct trajectory clustering at every MARL training step, we introduce a classifier to determine the class of the newly obtained trajectories from the environment.

- **Trajectory-class-aware policy:** After identifying the trajectory class, we train the agentwise trajectory-class predictor, which predicts a trajectory class using agents' partial observations. Using this prediction, the trajectory-class representation model generates a trajectory-class-dependent representation. This task type or trajectory-class representation is then provided to an action policy along with local observations. In this way, agents can learn trajectory-class-dependent policy.

Figure 1 illustrates the conceptual process of TRAMA by enumerating key functionalities on integrating the trajectory-class information in decision-making.

## 2 PRELIMINARIES

### 2.1 DECENTRALIZED POMDP

Decentralized Partially Observable Markov Decision Process (Dec-POMDP) (Oliehoek & Amato, 2016) is a widely adopted formalism for general cooperative multi-agent reinforcement learning (MARL) tasks. In Dec-POMDP, we define the tuple $G = \langle I, S, \mathcal{A}, P, R, \mathbf{\Omega}, O, n, \gamma \rangle$, where $I$ is the finite set of $n$ agents; $s \in S$ is the true state of the global state space $S$; $\mathcal{A} = \times_i \mathcal{A}^i$ is the joint action space and the joint action $\boldsymbol{a}$ is formed by each agent's action $a^i \in \mathcal{A}^i$; $P(s'|s, \boldsymbol{a})$ is the state transition function to new state $s' \in S$ given $s$ and $\boldsymbol{a}$; a reward function $R$ provides a scalar reward $r = R(s, \boldsymbol{a}, s') \in \mathbb{R}$ to a given transition $\langle s, a \rangle \to s'$; $O$ is the observation function generating each

agent's observation $o^i \in \Omega^i$ from the joint observation space $\boldsymbol{\Omega} = \times_i \Omega^i$; and finally, $\gamma$ is a discount factor. At each timestep, an agent receives a local observation $o^i$ and takes an action $a^i \in \mathcal{A}^i$ given $o^i$. Given the global state $s$ and joint action $\boldsymbol{a}$, state transition function $P(s'|s, \boldsymbol{a})$ determines the next state $s'$. Then, $R$ provides a common reward $r = R(s, \boldsymbol{a}, s')$ to all agents. For MARL training, we follow the conventional value factorization approaches under the CTDE framework. Please refer to Appendix A.3 for details.

## 2.2 Multi-agent Multi-Task

This section introduces a formal definition of a multi-agent multi-task $\mathcal{T}$, under dec-POMDP settings. In this paper, we omit the term multi-agent and denote *multi-task* for conciseness.

**Definition 2.1** *(Multi-agent multi-task $\mathcal{T}$) A partially observable multi-agent multi-task $\mathcal{T}$ is defined by a tuple $\langle I, S, \boldsymbol{\mathcal{A}}, P, R, \boldsymbol{\Omega}, O, n, \gamma, \mathcal{K} \rangle$, where $\mathcal{K}$ is a set of tasks, $S = \cup_k S_k$ and $\boldsymbol{\Omega} = \cup_k \boldsymbol{\Omega}_k$ for task-specific state space $S_k$ and joint observation space $\Omega_k$. Then, a partially-observable single-task $\mathcal{T}_k$ for $k \in \mathcal{K}$ is defined by a tuple of $\langle I, S_k, \boldsymbol{\mathcal{A}}, P, R, \boldsymbol{\Omega}_k, O, n, \gamma \rangle$, such that $\forall k_1, k_2 \in \mathcal{K}$, $S_{k_1} \cap S_{k_2}^c \neq \emptyset$ and $\boldsymbol{\Omega}_{k_1} \cap \boldsymbol{\Omega}_{k_2}^c \neq \emptyset$.*

**Support Difference** Although each task $\mathcal{T}_k$ shares the governing transition $P$, reward $R$, and observation $O$ functions, Definition 2.1 implies that $\forall k_1, k_2 \in K$, $\mathrm{dom}(P)_{k_1} \cap \mathrm{dom}(P)_{k_2}^c \neq \emptyset$, $\mathrm{dom}(R)_{k_1} \cap \mathrm{dom}(R)_{k_2}^c \neq \emptyset$ and $\mathrm{dom}(O)_{k_1} \cap \mathrm{dom}(O)_{k_2}^c \neq \emptyset$, where subscript $k_1$ and $k_2$ represent task specific values. For example, two tasks with different unit combinations in SMACv2 (Ellis et al., 2024) satisfy this condition. Figure 2 illustrates the state diagram of multi-task settings. Thus, agents need to learn generalizable policies to maximize the expected return obtained from $\mathcal{T}$.

**Unsupervised Multi-Task** Importantly, in multi-task setting in this paper, *task ID $k$ is unknown in both training and execution.* This differs from general multi-task learning, where task ID is generally given during training (Omidshafiei et al., 2017; Hansen et al., 2024; Yu et al., 2020; Tassa et al., 2018). This unsupervised multi-task setting is practical for multi-agent tasks. For example, in a football game, allied teammates predict opponents' strategies based on their observations during competition to respond appropriately. In such cases, a task label indicating the opponents' strategy (or task type) is not provided to the allied team or cooperating agents.

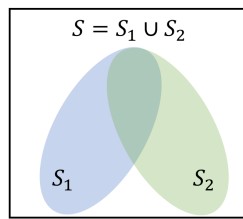

Figure 2: State Diagram of multi-task setting

This setting can also be viewed as a formal definition of SMACv2 task (Ellis et al., 2024). To address this challenging multi-task problem, TRAMA begins by identifying which task a given trajectory belongs to through clustering, assuming that trajectories from the same task are more similar than those from different tasks.

## 2.3 Quantized Latent Space Generation with VQ-VAE

In this paper, we utilize VQ-VAE (Van Den Oord et al., 2017) to generate trajectory embeddings in a discretized latent space. We follow VQ-VAE adoption in MARL introduced by LAGMA (Na & Moon, 2024). The VQ-VAE for state embedding in MARL contains an encoder network $f_\phi^e : S \to \mathbb{R}^d$, a decoder network $f_\phi^d : \mathbb{R}^d \to S$, and trainable embedding vectors used as codebook with size $n_c$ denoted by $\boldsymbol{e} = \{e_1, e_2, ...e_{n_c}\}$ where $e_j \in \mathbb{R}^d$ for all $j = \{1, 2, ..., n_c\}$. An encoder output $x = f_\phi^e(s) \in \mathbb{R}^d$ is replaced to discretized latent $x_q$ by *quantization process* $[\cdot]_q$, which maps $x$ to the nearest embedding vector in codebook $\boldsymbol{e}$ as follows.

$$x_q = [x]_q = e_z, \text{where } z = \mathrm{argmin}_j ||x - e_j||_2 \tag{1}$$

Then, a decoder $f_\phi^d$ reconstructs the original state $s$ given quantized vector input $x_q$. We follow the objective presented by LAGMA to train an encoder $f_\phi^e$, a decoder $f_\phi^d$, and codebook $\boldsymbol{e}$.

$$\mathcal{L}_{VQ}^{tot}(\phi, \boldsymbol{e}) = \mathcal{L}_{VQ}(\phi, \boldsymbol{e}) + \lambda_{\mathrm{cvr}} \frac{1}{|\mathcal{J}(t)|} \sum_{j \in \mathcal{J}(t)} ||\mathrm{sg}[f_\phi^e(s)] - e_j||_2^2 \tag{2}$$

$$\mathcal{L}_{VQ}(\phi, \boldsymbol{e}) = ||f_\phi^d([f_\phi^e(s)]_q) - s||_2^2 + \lambda_{\mathrm{vq}} ||\mathrm{sg}[f_\phi^e(s)] - x_q||_2^2 + \lambda_{\mathrm{commit}} ||f_\phi^e(s) - \mathrm{sg}[x_q]||_2^2 \tag{3}$$

Here, sg[·] represents a stop gradient. $\lambda_{\text{vq}}$, $\lambda_{\text{commit}}$, and $\lambda_{\text{cvr}}$ are scale factors for corresponding terms. We follow a straight-through estimator to approximate the gradient signal for an encoder (Bengio et al., 2013). The last term in Eq. (2) is a *coverage loss* to spread the quantized vectors throughout the embedding space. $\mathcal{J}(t)$ is a timestep-dependent index, which designates some portion of quantized vectors to a given timestep. Although the previous coverage loss works well in general MARL tasks, it is observed that such $\mathcal{J}(t)$ has limitations in multiple tasks. We modify this $\mathcal{J}(t)$ by identifying types of trajectories during the training. As a result, we expand the index to include timestep and task type, and thus the indexing function is now $\mathcal{J}(t, k)$, incorporating both timestep $t$ and trajectory class $k$.

## 3 METHODOLOGY

This section presents **TR**ajectory-class-**A**ware **M**ulti-**A**gent reinforcement learning (**TRAMA**). To learn trajectory-class-dependent policy, we first generate trajectory embeddings before performing trajectory clustering. To this end, we construct a **(1) quantized latent space** using modified VQ-VAE. With the trajectory embeddings in quantized latent space, we then describe the process for **(2) trajectory clustering and trajectory classifier learning**. Finally, we present the **(3) trajectory-class-aware policy**, which consists of a trajectory-class predictor and a trajectory-class representation model, in addition to the action policy network.

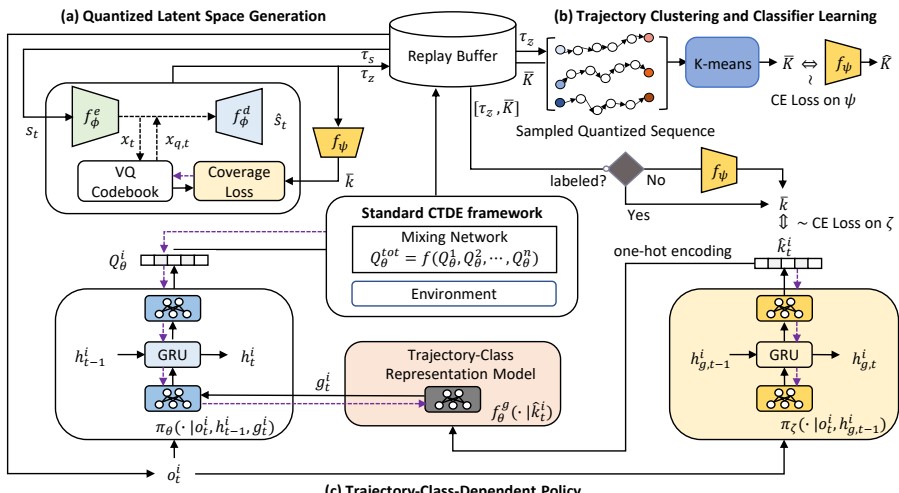

Figure 3: Overview of TRAMA framework. The purple dashed line represents a gradient flow.

### 3.1 QUANTIZED LATENT SPACE GENERATION WITH MODIFIED VQ-VAE

This paper adopts VQ-VAE (Van Den Oord et al., 2017) for trajectory embedding in quantized latent space. In this way, trajectory embedding can be represented by the sequence of quantized vectors. As illustrated in Section 2.3, LAGMA (Na & Moon, 2024) presents the coverage loss utilizing timestep dependent indexing $\mathcal{J}(t)$ to distribute quantized vector evenly throughout the embedding space of states stored in the current replay buffer $\mathcal{D}$, denoted as $\chi = \{x \in \mathbb{R}^d : x = f_\phi^e(s), s \in \mathcal{D}\}$. However, in multi-task settings, state distributions are different according to tasks, so $\mathcal{L}_{\text{cvr}}$ with $\mathcal{J}(t)$ does not guarantee quantized vectors evenly distributed over $\chi$. To resolve this, we additionally consider the trajectory class $k$ in the coverage loss through a modified indexing function, $\mathcal{J}(t, k)$, which designates specific quantized vectors in the codebook, according to $(t, k)$ pair. When the class of a given trajectory $\tau_{s_{t=0}} = \{s_{t=0}, s_{t=1}, \cdots, s_{t=T}\}$ is identified as $k$, we consider a state $s_t \in \tau_{s_t}$ to belong to the $k$-th class and denote this state as $s_t^k$. Then, the modified coverage loss, considering both timestep $t$ and the trajectory class $k$, is expressed as follows.

$$\mathcal{L}_{\text{cvr}}(e) = \frac{1}{|\mathcal{J}(t, k)|} \sum_{j \in \mathcal{J}(t,k)} ||\text{sg}[f_\phi^e(s_t^k)] - e_j||_2^2 \tag{4}$$

Eq. (4) adjusts quantized vectors assigned to the $k$-th class towards embedding of $s^k$. The details

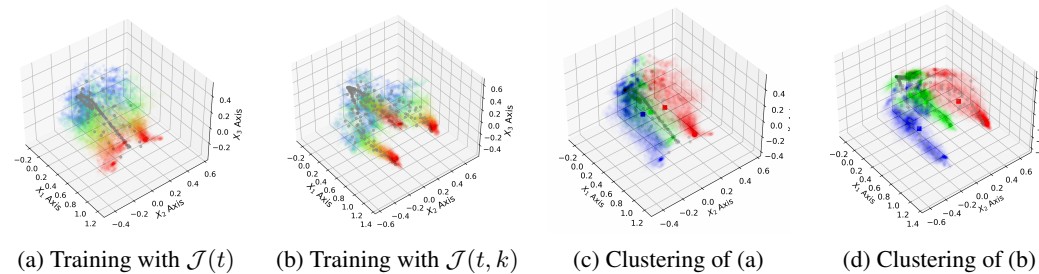

(a) Training with $\mathcal{J}(t)$  (b) Training with $\mathcal{J}(t,k)$  (c) Clustering of (a)  (d) Clustering of (b)

Figure 4: PCA of sampled embedding $x \in \mathcal{D}$. Colors from red to purple (rainbow) represent early to late timestep in (a) and (b). (a) and (b) are the results of sample multi-tasks with three different unit combinations with various initial positions. (c) and (d) are the clustering results of (a) and (b), respectively, and each color (red, green, and blue) represents each class.
Here, the class number $n_{cl} = 3$ is assumed.

of $\mathcal{J}(t,k)$ are presented in Appendix C. Then, with a given $s_t^k$, the final loss function $\mathcal{L}_{VQ}^{tot}$ to train VQ-VAE becomes

$$\mathcal{L}_{VQ}^{tot}(\phi, \boldsymbol{e}) = \mathcal{L}_{VQ}(\phi, \boldsymbol{e}) + \lambda_{\text{cvr}} \frac{1}{|\mathcal{J}(t,k)|} \sum_{j \in \mathcal{J}(t,k)} ||\text{sg}[f_\phi^e(s_t^k)] - e_j||_2^2. \tag{5}$$

Figure 4 illustrates the embedding results with the proposed coverage loss compared to the original $\mathcal{J}(t)$. As in Figure 4 (b), the proposed coverage loss $\mathcal{J}(t,k)$ distributes quantized vectors more evenly throughout $\chi$ in multiple tasks compared to the cases with $\mathcal{J}(t)$ in (a). Notably, we assumed the number of classes as $n_{cl} = 6$ in (b). Still, our model successfully captured three distinct initial unit combinations in the task, as illustrated by the three red branches in Figure 4 (b). However, to adopt this modified coverage loss, we need to determine which trajectory class a given state belongs to. Thus, we conduct clustering to annotate the trajectory class. Then, we use pseudo-class $\bar{k}$ obtained from clustering for Eq. (5).

## 3.2 TRAJECTORY CLUSTERING AND CLASSIFIER LEARNING

**Trajectory Clustering** With a given trajectory $\tau_{s_{t=0}}$, we get a quantized latent sequence with VQ-VAE as $\tau_{\chi_{t=0}} = [f_\phi^e(\tau_{s_t})]_q = \{x_{q,t=0}, x_{q,t=1}, \cdots, x_{q,t=T}\}$. Here, only the indices of quantized vectors, i.e., $\tau_{\mathcal{Z}_{t=0}} = \{z_{t=0}, z_{t=1}, \cdots, z_{t=T}\}$, are required to express $\tau_{\chi_{t=0}}$. Thus, we can efficiently store the quantized sequence $\tau_{\mathcal{Z}_{t=0}}$ to $\mathcal{D}$ along with the given trajectory, $\tau_{s_{t=0}}$. In addition to MARL training, we sample $M$ trajectory sequences $[\tau_{\mathcal{Z}_{t=0}}^m]_{m=1}^M$ from $\mathcal{D}$ and conduct *K-means clustering* (Lloyd, 1982; Arthur & Vassilvitskii, 2006) periodically.

With the $m$-th index sequence $\tau_{\mathcal{Z}_{t=0}}^m$, we compute a trajectory embedding $\bar{e}^m$ using quantized vectors $\boldsymbol{e}$ in the codebook.

$$\bar{e}^m = \sum_{t=0}^T e_{j=z_t}^m \tag{6}$$

Then, with trajectory embeddings $[\bar{e}^m]_{m=1}^M$, we conduct K-means clustering with the predetermined number of class $n_{cl}$. In this paper, the class labels are denoted as $\bar{K} = \{\bar{k}_{m=1}, \bar{k}_{m=2}, ..., \bar{k}_{m=M}\}$. Figures 4 (c) and (d) illustrate the clustering results with trajectory

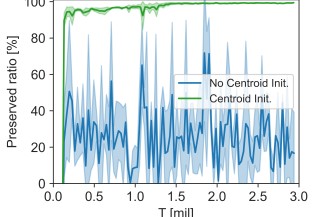

Figure 5: Preserved Labels

embedding constructed by Eq. 6 based on quantized vectors of (a) and (b), respectively. The visual results and their silhouette score emphasize the importance of the distribution of quantized vectors. Appendix D.4 provides further analysis. However, the problem is that the class labels may change whenever the clustering is updated. Consistent class labels are important because we update the agent-wise predictor based on these labels. To resolve this, we conduct *centroid initialization* with the previous centroid results. Figure 5 shows the ratio of preserved labels after clustering with and without considering centroid initialization.

**Trajectory Classifier Training** Even though we determine trajectory-class labels $\bar{K}$ stored in $\mathcal{D}$ at a specific training time, we do not have labels for new trajectories obtained by interacting with the

environment. To determine the labels for such trajectories before additional clustering update, we develop a classifier $f_\psi(\cdot|\bar{e}^m)$ to predict a trajectory-class $\hat{k}_m$ based on $\bar{e}^m$. We train $f_\psi$ whenever clustering is updated in parallel to MARL training through cross-entropy loss, $\mathcal{L}_\psi$ with $M$ samples.

$$\mathcal{L}(\psi) = -\frac{1}{M} \sum_{m=1}^{M} \mathbf{1}_{\bar{k}_m = \hat{k}_m} \log(f_\psi(\hat{k}_m|\bar{e}^m)) \tag{7}$$

Here, $\mathbf{1}$ is an indicator function. Figure 6 illustrates the loss classifier as training proceeds. With centroid initialization, the classifier learns trajectory embedding patterns more coherently than the case without it.

### 3.3 Trajectory-Class-Aware Policy

**Trajectory-Class Predictor** After obtaining the class labels $\bar{K}$ for sampled trajectories either determined by clustering or a classifier $f_\psi$, we can train a *trajectory-class predictor* $\pi_\zeta$ shared by all agents. Unlike the trajectory classifier $f_\psi$, a trajectory-class predictor $\pi_\zeta$ only utilizes partial observation given to each agent. In other words, each agent makes a prediction on which trajectory type or class it is experiencing based on its partial observation, such as $\hat{k}_t^i \sim \pi_\zeta(\cdot|h_{g,t}^i)$. Here, $h_{g,t}^i$ represents the observation history computed by GRUs in $\pi_\zeta$. With predetermined trajectory labels for sampled batches size of $B$, we train $\pi_\zeta$ with the following loss.

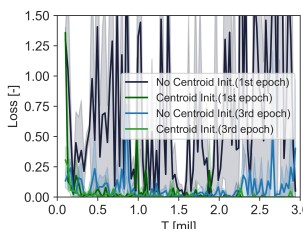

Figure 6: Classifier Loss

$$\mathcal{L}(\zeta) = -\frac{1}{B} \sum_{b=1}^{B} \left[ \Big[ \sum_{t=0}^{T-1} \sum_{i=1}^{n} \mathbf{1}_{\bar{k}=\hat{k}_t^i} \log(\pi_\zeta(\hat{k}_t^i|o_t^i, h_{g,t}^i)) \Big] \right]_b \tag{8}$$

**Trajectory-Class Representation Learning** Base on $\pi_\zeta$, each agent predicts $\hat{k}_t^i$ at each timestep $t$. Then, the $i$-th agent utilizes one-hot encoding of $\hat{k}_t^i$ as an additional condition or prior when determining its action. Instead of directly utilizing this one-hot vector, we train a *trajectory-class representation model* $f_\theta^g(\cdot|\hat{k}_t^i)$ to generate a more informative representation, $g_t^i$. The purpose of $g_t^i$ is to generate coherent information for decision-making and to enable agents to learn a trajectory-class-dependent policy. As illustrated in Figure 3, we directly train $f_\theta^g$ through MARL training. In addition, we use a separate network for action policy $\pi_\theta$ and utilize the additional class representation $g_t^i$ along with partial observation $o_t^i$ for decision-making.

### 3.4 Overall Learning Objective

This paper adopts value function factorization methods (Rashid et al., 2018; Wang et al., 2020a; Rashid et al., 2020; Zheng et al., 2021) presented in Section A.3 to train individual $Q_\theta^i$ via $Q_\theta^{tot}$. For a mixer structure, we mainly adopt QPLEX (Wang et al., 2020a), which guarantees the complete Individual-Global-Max (IGM) condition (Son et al., 2019). The loss function for the action policy $Q_\theta^i$ and $f_\theta^g$ can be expressed as

$$\mathcal{L}(\theta) = \mathbb{E}_{k \sim p(k)} \big[ \mathbb{E}_{\langle \boldsymbol{o}, \boldsymbol{a}, r, \boldsymbol{o}' \rangle_k \sim \mathcal{D}, \hat{k} \sim \pi_\zeta(\cdot|o), g \sim f_\theta^g(\cdot|\hat{k})} [(r + \gamma \max_{\boldsymbol{a}'} Q_{\theta-}^{tot}(\boldsymbol{o}', \boldsymbol{g}', \boldsymbol{a}') - Q_\theta^{tot}(\boldsymbol{o}, \boldsymbol{g}, \boldsymbol{a}))^2]\big].$$

$$= \mathbb{E}_{\boldsymbol{o}, \boldsymbol{a}, r, \boldsymbol{o}' \sim \mathcal{D}, \hat{k} \sim \pi_\zeta(\cdot|o), g \sim f_\theta^g(\cdot|\hat{k})} [(r + \gamma \max_{\boldsymbol{a}'} Q_{\theta-}^{tot}(\boldsymbol{o}', \boldsymbol{g}', \boldsymbol{a}') - Q_\theta^{tot}(\boldsymbol{o}, \boldsymbol{g}, \boldsymbol{a}))^2]. \tag{9}$$

Here, $p(k)$ is the portion of samples $\langle \boldsymbol{o}, \boldsymbol{a}, r, \boldsymbol{o}' \rangle_k$ generated by $\mathcal{T}_k$ within $\mathcal{D}$. However, since we randomly sample a tuple from $\mathcal{D}$, the expectation over $k$ can be omitted. In addition, $\boldsymbol{g}$ represents the joint trajectory-class representation. With this loss function, we train $Q_\theta^i$, $f_\theta^g$, and $\pi_\zeta$ together with the following learning objective:

$$\mathcal{L}(\theta, \zeta) = \mathcal{L}(\theta) + \lambda_\zeta \mathcal{L}(\zeta). \tag{10}$$

Here, $\lambda_\zeta$ is a scale factor. Note that $\theta$ denotes neural network parameters contained in both $Q_\theta$ and $f_\theta^g$. Algorithm 2 in Appendix C specifies the learning procedure with loss functions specified by Eqs. (5), (7), (8) and (10).

# 4 RELATED WORKS

## 4.1 TASK DIVISION METHODS IN MARL

In the field of MARL, task division methods are introduced in diverse frameworks. Although previous works use different terminology, such as a subtask (Yang et al., 2022), role (Wang et al., 2020b; 2021) or skill (Yang et al., 2019; Liu et al., 2022), they share the primary objective, such as reducing search space during training or encouraging committed behaviors among agents utilizing conditioned policies. HSD (Yang et al., 2019), RODE (Wang et al., 2020b), LDSA (Yang et al., 2022) and HSL (Liu et al., 2022) adopt a hierarchical structure where upper-tier policy network first determines agents' roles, skills, or subtasks, and then agents determine actions based on these additional conditions along with their partial observations. These approaches share a commonality with goal-conditioned RL in single-agent tasks; however, the major difference is that the goal is not explicitly defined in MARL. MASER (Jeon et al., 2022) adopts a subgoal generation scheme from goal-conditioned RL when it generates an intrinsic reward. On the other hand, TRAMA first clusters trajectories considering their commonality among multi-tasks. Then, agents predict task types by identifying the trajectory class and use this prediction as additional information for action policy. In this way, agents utilize trajectory-class-dependent or task-specific policies.

Appendix A presents additional related works regarding state space abstraction and some prediction methods developed for MARL.

# 5 EXPERIMENTS

In this section, we evaluate TRAMA through multi-task problems built upon SMACv2 (Ellis et al., 2024) and conventional MARL benchmark problems (Samvelyan et al., 2019; Ellis et al., 2024). We have designed the experiments to observe the following aspects.

- Q1. The performance of TRAMA in multi-task problems and conventional benchmark problems compared to state-of-the-art MARL frameworks
- Q2. The impact of the major components of TRAMA on agent-wise trajectory-class prediction and overall performance
- Q3. Trajectory-class distribution in the embedding space

To compare the performance of TRAMA, we consider various baseline methods: popular baseline methods such as QMIX (Rashid et al., 2018) and QPLEX (Wang et al., 2020a); subtask-based methods such as RODE (Wang et al., 2021), LDSA (Yang et al., 2022), and MASER (Jeon et al., 2022); memory-based approach such as EMC (Zheng et al., 2021) and LAGMA (Na & Moon, 2024). For the baseline methods, we follow the hyperparameter settings presented in their original paper and implementation. For TRAMA, the details of hyperparameter settings are presented in Appendix B.

## 5.1 COMPARATIVE EVALUATION ON BENCHMARK PROBLEMS

We first evaluate TRAMA on the conventional SMACv2 tasks (multi-tasks) such as p5_vs_5 and t5_vs_5 in (Ellis et al., 2024). In Figure 7, TRAMA shows better learning efficiency and performance compared to other baseline methods, including a memory-based approach such as EMC (Zheng et al., 2021), utilizing an additional memory buffer. Besides multi-task problems, we conduct additional experiments on the original SMAC (Samvelyan et al., 2019) tasks

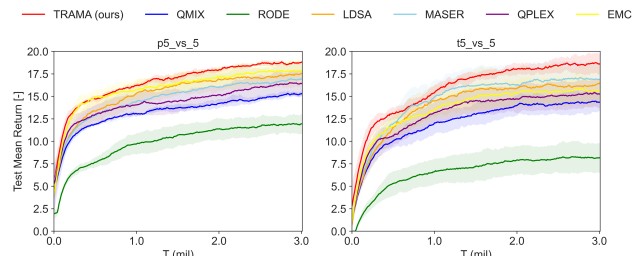

Figure 7: Performance comparison of TRAMA against baseline algorithms on p5_vs_5 and t5_vs_5 in SMACv2 (multi-tasks). Here, $n_{cl}$=8 is assumed.

to see how TRAMA works in single-task settings. Figure 8 illustrates the results, and TRAMA shows comparable or better performance compared to other methods. Notably, TRAMA succeeds

in learning the best policy at the end in super hard tasks such as `MMM2` and `6h_vs_8z`. In the following section, we present a parametric study on $n_{cl}$ and explain how we determine the appropriate value based on the tasks.

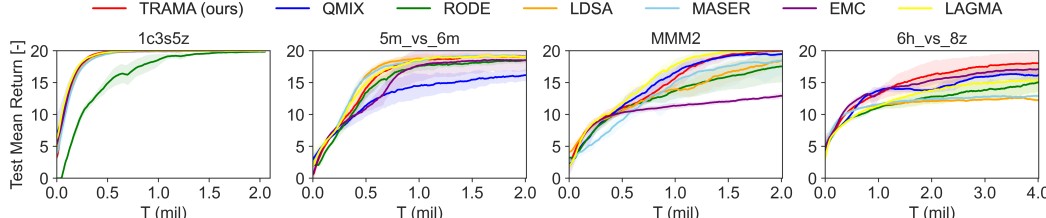

Figure 8: Performance comparison of TRAMA compared to baseline algorithms on SMAC task (single-task).

## 5.2 COMPARATIVE EVALUATION ON VARIOUS MULTI-TASK PROBLEMS

To test MARL algorithms on additional multi-task problems, we introduce the four modified tasks built upon SMACv2 as presented in Table 1. In these new tasks, two types of initial position distributions are considered, and initial unit combinations are randomly selected from the designated sets. In addition, agents get rewards only when each enemy unit is fully neutralized, similar to sparse reward settings in (Jeon et al., 2022; Na & Moon, 2024). Appendix B provides further details of multi-task problems and SMACv2.

Table 1: Task configuration of multi-task problems

| Name | Initial Position Type | Unit Combinations ($n_{comb}$) |
|---|---|---|
| SurComb3 | Surrounded | $\{3s2z, 2c3z, 2c3s\}$ |
| reSurComb3 | Surrounded and Reflected | $\{3s2z, 2c3z, 2c3s\}$ |
| SurComb4 | Surrounded | $\{1c2s2z, 3s2z, 2c3z, 2c3s\}$ |
| reSurComb4 | Surrounded and Reflected | $\{1c2s2z, 3s2z, 2c3z, 2c3s\}$ |

To evaluate performance, we consider the overall return value instead of the win-rate, as the learned policy may be specialized for specific tasks while being less effective for others among multiple tasks. Figure 9 illustrates the overall return values for multi-task problems. As illustrated in Figure 9, TRAMA consistently demonstrates better performance compared to other baseline methods.

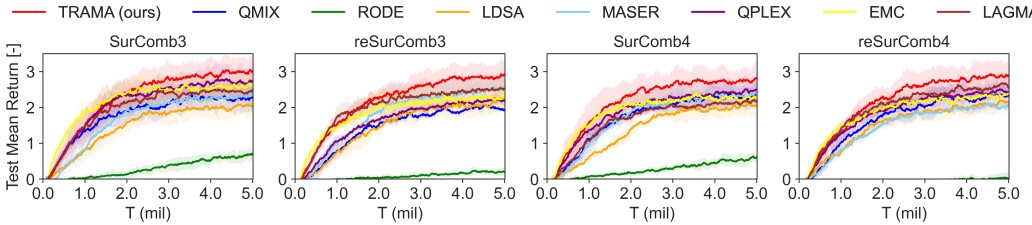

Figure 9: The mean return of TRAMA compared to baseline algorithms on four multi-task problems presented in Table 1.

To see how well each agent predicts the trajectory class, we present the learning loss of $\mathcal{L}_\zeta$ and the overall accuracy of the prediction on `reSurComb4` task in Figure 10. Appendix D.7 presents an additional analysis for a trajectory-class prediction made by agents. Notably, the agents accurately identify which types of trajectory classes they are experiencing based on their partial information throughout the episodes. In this case, agents can coherently generate trajectory-class representation $g$ through $f_\theta^g$ and condition on this additional prior information for decision-making. With this extra information, agents can learn distinct policies based on trajectory class and execute different joint policies specialized for each task, resulting in improved performance. In Section 5.5, we will discuss how trajectories are divided into different classes and represented in the embedding space.

## 5.3 PARAMETRIC STUDY

In this study, we check the performance variation according to key parameter $n_{cl}$ to evaluate the impact of the number of trajectory classes on the general performance. We considered $n_{cl} = \{2, 4, 8, 16\}$ for multi-task problems SurComb3 and SurComb4, and $n_{cl} = \{4, 6, 8, 16\}$ for the original SMACv2 task p5_vs_5. Figure 11 presents the overall return according to different $n_{cl}$. To evaluate the efficiency of training and performance together, we compare *cumulative return*, $\bar{\mu}_R$, which measures the area below the mean return curve. The high value of $\bar{\mu}_R$ represents better performance. In Figure 11 (d), $\bar{\mu}_R$ is normalized by its possible maximum value. From Figure 11 (d), we can see that peaks of $\bar{\mu}_R$

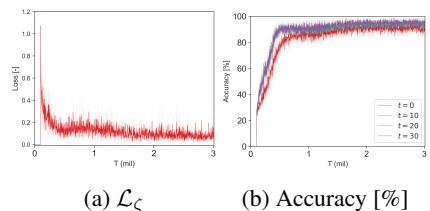

(a) $\mathcal{L}_\zeta$      (b) Accuracy [%]

Figure 10: With $n_{cl}$=4, learning loss of $\pi_\zeta$ and the mean accuracy of the trajectory-class prediction made by agents in (reSurComb4).

occur around $n_{cl} = 4$ for SurComb3 and SurComb4, and $n_{cl} = 8$ for p5_vs_5, respectively. Interestingly, these numbers seem highly related to variations in unit combinations. In SurComb3

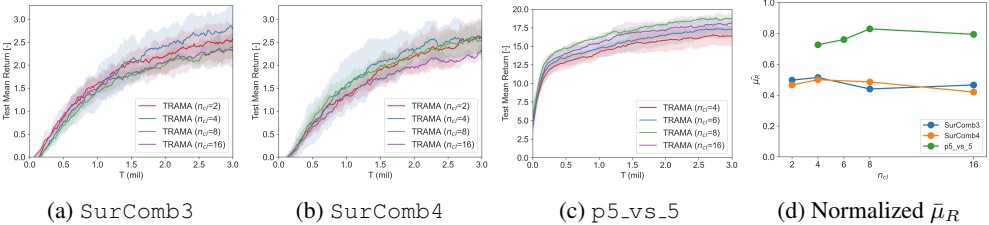

(a) SurComb3     (b) SurComb4     (c) p5_vs_5     (d) Normalized $\bar{\mu}_R$

Figure 11: Parametric study of $n_{cl}$ on surComb3, surComb4 and p5_vs_5.

and SurComb4, 3 and 4 unit combinations are possible, and thus $n_{cl} = 4$ well captures the trajectory diversity. Therefore, it is recommended to determine $n_{cl}$ considering the diversity of unit combinations of multi-tasks. When a larger $n_{cl}$ is selected, the agent-wise prediction accuracy may degrade because the number of options increases. However, *some classes share key similarities even though they are labeled as different classes*. Trajectory embedding in quantized embedding space can efficiently capture these similarities. Thus, agents can still learn coherent policies specialized on each task in multi-task settings even with different trajectory class labels. We will further elaborate on this in Section 5.5.

## 5.4 ABLATION STUDY

In this subsection, we conduct the ablation study to see the effect of major components of TRAMA. First, to see the importance of coherent label generation in class clustering, we consider **No-Init** representing clustering without centroid initialization presented in Section 3.2. In addition, we ablate the proposed coverage loss $\mathcal{J}(t, k)$ and consider $\mathcal{J}(t)$ instead when constructing quantized embedding space. Figure 12 illustrates the corresponding results.

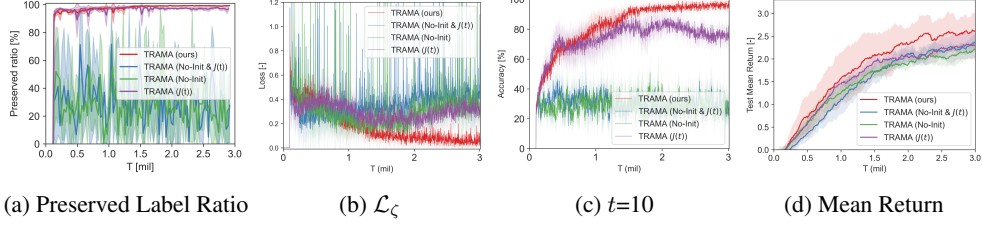

(a) Preserved Label Ratio     (b) $\mathcal{L}_\zeta$     (c) $t$=10     (d) Mean Return

Figure 12: Ablation study on SurComb4.

In Figure 12 (a), without centroid initialization, the trajectory class labels change frequently, and the loss for $\pi_\zeta$ fluctuates as illustrated in (b). As a result, $\pi_\zeta$ in both TRAMA (No-Init) and TRAMA (No-Init & $\mathcal{J}(t)$) predicts trajectory class labels almost randomly as illustrated in Figure 12 (c), leading to degraded performance as shown in Figure 12 (d). In the case of TRAMA ($\mathcal{J}(t)$), coherent

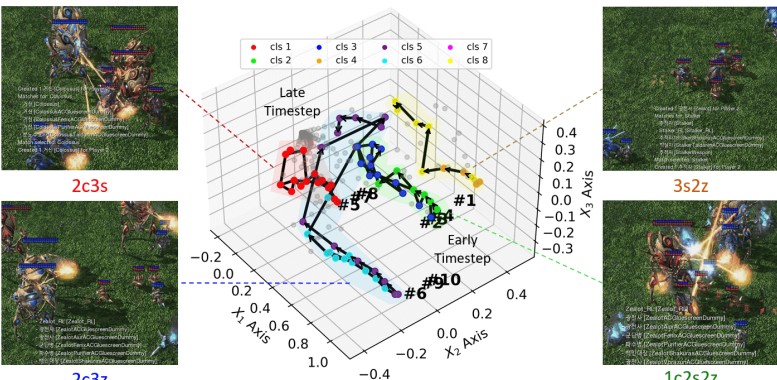

Figure 13: Visualization of embedding results and test episodes for `SurComb4`. Here, $n_{cl}=8$ is assumed, and gray dots represent quantized vectors in the VQ codebook. Solid lines represent each test episode, while colored dots represent the majority opinion on the trajectory class predictions made by agents. Each color denotes a different class.

labels are generated due to centroid initialization. However, prediction accuracy is lower and fluctuates compared to the full version of TRAMA, as the quantized vectors are not evenly distributed over $\chi$ as described in Figure 4. Therefore, TRAMA without $\mathcal{J}(t, k)$ cannot sufficiently capture the key differences in trajectories. In Appendix D, we present additional ablation studies regarding the class representation model $f_\theta^g$.

## 5.5 QUALITATIVE ANALYSIS

In this section, we evaluate how the trajectory classes are identified in the quantized embedding space. We consider `SurComb4` task as a test case and assume $n_{cl} = 8$ for this test. Figure 13 illustrates the visualization of embedding results and test episodes. Notably, four branches are developed in quantized embedding space after training, and each branch is highly related to the initial unit combinations. The result implies that the initial unit combinations significantly influence the trajectories of agents. Although the larger number of $n_{cl}$ is chosen compared to the number of possible unit combinations ($n_{comb} = 4$) in `SurComb4`, two classes are assigned to each branch in the quantized embedding space, making the model less sensitive to misclassification within each pair. In Figure 13, classes 4 and 8 are assigned to `3s2z`; classes 2 and 3 to `1c2s2z`; classes 5 and 6 to `2c3z`; and classes 1 and 7 to `2c3s`. By identifying the trajectory class, agents can generate additional prior information and utilize trajectory-class-dependent policies conditioned on this prediction. As we can see, this trajectory-class identification is important in solving multi-task problems $\mathcal{T}$. Thus, it would be interesting to see how TRAMA predicts trajectory classes in out-of-distribution tasks. As TRAMA learns to identify trajectory class in an unsupervised manner, without task ID, agents can identify similar tasks among in-distribution tasks. Then, agents rely on these predictions during decision-making, thereby promoting a joint policy that benefits OOD tasks. Appendix D.6 presents OOD experiments and their corresponding qualitative analysis, demonstrating TRAMA's generalizability across various OOD tasks.

## 6 CONCLUSION

This paper presents TRAMA, a new framework that enables agents to recognize task types by identifying the class of trajectories and to use this information for action policy. TRAMA introduces three major components: 1) construction of quantized latent space for trajectory embedding, 2) trajectory clustering, and 3) trajectory-class-aware policy. The constructed quantized latent space allows trajectory embeddings to share the key commonality between trajectories. With these trajectory embeddings, TRAMA successfully divides trajectories into clusters with similar task types in multi-tasks. Then, with a trajectory-class predictor, each agent predicts which trajectory types agents are experiencing and uses this prediction to generate trajectory-class representation. Finally, agents learn trajectory-class-aware policy with this additional information. Experiments validate the effectiveness of TRAMA in identifying task types in multi-tasks and in overall performance.

ACKNOWLEDGEMENT

This work was supported by the IITP(Institute of Information & Communications Technology Planning & Evaluation)-ITRC(Information Technology Research Center) grant funded by the Korea government(Ministry of Science and ICT)(IITP-2025-RS-2024-00437268).

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

## A  ADDITIONAL RELATED WORKS AND PRELIMINARIES

### A.1  STATE SPACE ABSTRACTION

It has been effective in grouping similar characteristics into a single cluster, which is called *state space abstraction*, in various settings such as model-based RL (Jiang et al., 2015; Zhu et al., 2021; Hafner et al., 2020) and model-free settings (Grześ & Kudenko, 2008; Tang & Agrawal, 2020). NECSA, introduced by (Li et al., 2023), is the model that facilitates the abstraction of grid-based state-action pairs for episodic control, achieving state-of-the-art (SOTA) performance in a general single-reinforcement learning task. The approach could alleviate the usage of inefficient memory in conventional episodic control. However, an additional dimensionality reduction process is inevitable, such as random projection in high-dimensional tasks in (Dasgupta, 2013). In (Na et al., 2024), EMU utilizes a state-based semantic embedding for efficient memory utilization. LAGMA (Na & Moon, 2024) employs VQ-VAE for state embedding and estimates the overall value of abstracted states to generate incentive structure encouraging transitions toward goal-reaching trajectories. Unlike previous works, we use state-space abstraction to generate trajectory embedding in quantized latent space, ensuring that these embeddings share key similarities. We then cluster the trajectories into several classes based on task commonality. In this way, TRAMA can learn a task-aware policy by identifying trajectory classes in multi-tasks.

### A.2  PREDICTIONS IN MARL

Predictions in MARL are generally used to model the actions of agents (Zhang & Lesser, 2010; He et al., 2016; Grover et al., 2018; Raileanu et al., 2018; Papoudakis et al., 2021; Yu et al., 2022). In (Carion et al., 2019; Li et al., 2020; Christianos et al., 2021), models also include groups or agents' tasks for their prediction. The authors in (He et al., 2016) utilize the Q-value to predict opponents' actions, assuming varying opponents' policies. On the other hand, Raileanu et al. (2018) presents the model to predict other agents' actions by updating hidden states. Papoudakis & Albrecht (2020) introduce the opponent modeling adopting variational autoencoder (VAE) and A2C without accessibility to opponents' information. LIAM (Papoudakis et al., 2021) learns the trajectories of the modeled agent using those of the controlled agent. On the other hand, in our approach, agents predict which trajectory class they are experiencing to generate additional inductive bias for decision-making based on this prediction.

### A.3  CENTRALIZED TRAINING WITH DECENTRALIZED EXECUTION (CTDE)

Under Centralized Training with Decentralized Execution (CTDE) framework, value factorization approaches have been introduced by (Sunehag et al., 2017; Rashid et al., 2018; Son et al., 2019; Rashid et al., 2020; Wang et al., 2020a) to solve fully cooperative multi-agent reinforcement learning (MARL) tasks, and these approaches achieved state-of-the-art performance in challenging benchmark problems such as SMAC (Samvelyan et al., 2019). Value factorization approaches utilize the joint action-value function $Q_\theta^{tot}$ with learnable parameter $\theta$. Then, the training objective $\mathcal{L}(\theta)$ can be expressed as

$$\mathcal{L}(\theta) = \mathbb{E}_{\boldsymbol{o},\boldsymbol{a},r,\boldsymbol{o}'\sim\mathcal{D}}[\left(r + \gamma\max_{\boldsymbol{a}'}Q_{\theta^-}^{tot}(\boldsymbol{o}',\boldsymbol{a}') - Q_\theta^{tot}(\boldsymbol{o},\boldsymbol{a})\right)^2], \tag{11}$$

where $\mathcal{D}$ is a replay buffer; $\boldsymbol{o}$ is the joint observation; $Q_{\theta^-}^{tot}$ is a target network with online parameter $\theta^-$ for double Q-learning (Hasselt, 2010; Van Hasselt et al., 2016); and $Q_\theta^{tot}$ and $Q_{\theta^-}^{tot}$ include both mixer and individual policy network.

## B  EXPERIMENT DETAILS

### B.1  EXPERIMENT DESCRIPTION

In this section, we present details of SMAC (Samvelyan et al., 2019), SMACv2 (Ellis et al., 2024) and multi-task problems presented in Table 1 built upon StarCraft II. To test the generalization of policy, SMACv2 contains highly varying initial positions and different unit combinations within one *map*, unlike the original SMAC tasks. In new tasks, agents may require different strategies against

enemies with different unit combinations and initial positions. TRAMA makes agents recognize which task they are solving and then utilize these predictions as additional conditions for action policy. Figure 14 compares the different characteristics between single-task and multi-tasks.

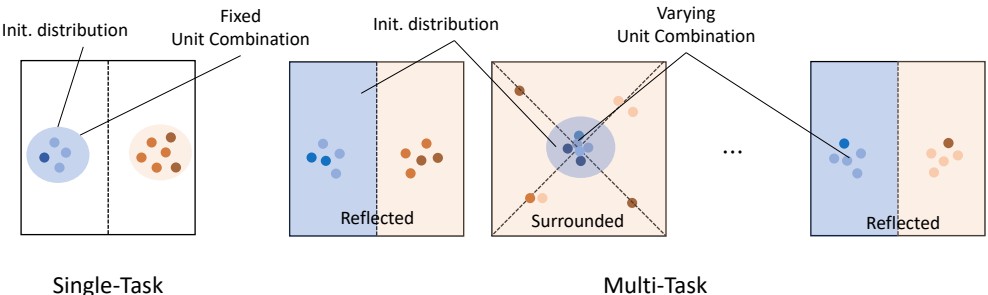

Figure 14: Comparison between single-task and multi-tasks.

In SMACv2, unit combinations for agents are randomly selected from a set of given units based on the predefined selection probability for each unit. For example, in `t5_vs_5`, five units are drawn from three possible units, such as `Marine`, `Marauder` and `Medivac`, according to predetermined probabilities. Initial unit positions are randomly selected between `Surrounded` and `Reflected`.

On the other hand, in our multi-tasks presented in Table 1, initial unit combinations are selected from the predefined sets. For example, three unit combinations are possible among {`3s2z`, `2c3z`, `2c3s`} in `SurComb3`. Here, s, z, and c represent `Stalker`, `Zealot`, and `Colossus`, respectively. In addition, in our multi-tasks, we adopt **sparse reward settings** similar to (Jeon et al., 2022; Na & Moon, 2024) unlike SMACv2. Table 2 presents the reward structure of the multi-tasks we presented.

Table 2: Reward settings for multi-tasks.

| Condition | Sparse reward |
|---|---|
| All enemies die (Win) | +200 |
| Each enemy dies | +10 |
| Each ally dies | -5 |

Note that both SMAC (Samvelyan et al., 2019) and SMACv2 (Ellis et al., 2024) normalize the reward output by the maximum reward agents can get so that the maximum return (without considering discount factor) becomes `20`. Due to this setting, the maximum return in our multi-tasks becomes around `3.5`. Table 3 presents details of tasks evaluated in the experiment section.

Table 3: Task Specification

| | Task | $n_{\text{agent}}$ | Dim. of state space | Dim. of action space | Episodic length |
|---|---|---|---|---|---|
| Multi-task | SurComb3 | 5 | 130 | 11 | 200 |
| | reSurComb3 | 5 | 130 | 11 | 200 |
| | SurComb4 | 5 | 130 | 11 | 200 |
| | reSurComb4 | 5 | 130 | 11 | 200 |
| SMACv2 | p5_vs_p5 | 5 | 130 | 11 | 200 |
| | t5_vs_t5 | 5 | 120 | 11 | 200 |
| SMAC | 1c3s5z | 9 | 270 | 15 | 180 |
| | 5m_vs_6m | 5 | 120 | 11 | 70 |
| | MMM2 | 10 | 322 | 18 | 180 |
| | 6h_vs_8z | 6 | 140 | 14 | 150 |

## B.2 Experiment Settings

For the performance evaluation, we measure the mean return computed with 128 samples: 32 episodes for four different random seeds. For baseline methods, we follow the settings presented in the original papers or their original codes. We use almost the same hyperparameters throughout the various tasks except for $n_{cl}$. For VQ-VAE training, we use the fixed hyperparameters for all tasks, such as $\lambda_{\text{vq}}$=0.25, $\lambda_{\text{commit}}$=0.125, $\lambda_{\text{cvr}}$=0.125 in Eq. (5), $n_\psi$=500, and $n_{\text{freq}}^{vq}$=10. Here, $n_\psi$ is the update interval for clustering and classifier learning, and $n_{\text{freq}}^{vq}$ represents the update interval of VQ-VAE. Algorithm 2 presents details of TRAMA training and the parameters used in overall training. Table 4 summarizes the task-dependent hyperparameter settings for TRAMA. Here, the dimension of latent space is denoted as $d$.

Table 4: Hyperparameter settings for TRAMA experiments.

|  | Task | $n_c$ | $d$ | $n_{cl}$ | $\epsilon_T$ |
|---|---|---|---|---|---|
| Multi-task | SurComb3 | 256 | 4 | 4 | 50K |
|  | reSurComb3 |  |  | 6 |  |
|  | SurComb4 |  |  | 4 |  |
|  | reSurComb4 |  |  | 8 |  |
| SMACv2 | p5_vs_p5 | 256 | 4 | 8 | 50K |
|  | t5_vs_t5 |  |  | 8 |  |
| SMAC | 1c3s5z | 256 | 4 | 3 | 50K |
|  | 5m_vs_6m |  |  | 3 |  |
|  | MMM2 |  |  | 3 |  |
|  | 6h_vs_8z |  |  | 4 | 200K |

## B.3 Infrastructure and Code Implementation

For experiments, we mainly use GeForce RTX 3090 and GeForce RTX 4090 GPUs. Our code is built on PyMARL (Samvelyan et al., 2019) and the open-sourced code from LAGMA (Na & Moon, 2024). Our official code is available at: https://github.com/aailab-kaist/TRAMA.

## B.4 Training and Computational Time Analysis

This section presents the total training time required for each model to learn each task. Before that, we present the computational costs of newly introduced modules in TRAMA. Table 5 presents the results. In Table 5, the MARL training module includes training for prediction and VQ-VAE modules. The computational cost is about 20% increased compared to the model without a prediction and VQ-VAE modules, overall training time does not increase much compared to other complex baseline methods as illustrated in Table 6. This is because most computational load in MARL often comes from rolling out sample episodes by interacting with the environment. In addition, as VQ-VAE module is periodically updated, we measure the mean computational time with VQ-VAE updates. Clustering and classifier training are called sparsely compared to MARL training; overall, additional computational costs are not burdensome. In addition, we can expedite computational costs for classifier training, as a classifier already converges to optimal one after sufficient training, as illustrated in Figure 6.

Table 5: Computational costs of TRAMA modules.

| Module | Computing time per call [s] |
|---|---|
| MARL training module | 0.83 |
| Clustering | 0.13~0.3 |
| Classifier training | 2~15 |

Training times of all models are measured in GeForce RTX 3090 or RTX 4090. In Table 6, marker (*) represents the training time measured by GeForce RTX 4090. Others are measured by RTX 3090. As in Table 6, TRAMA does not take much training time compared to other baseline methods, even with a periodic update for trajectory clustering, classifier learning, and VQ-VAE training. Again, as a classifier already converges to optimal one after sufficient training as illustrated in Figure 6, we can further expedite training speed by reducing the frequency of updating the classifier, $f_\psi$.

Table 6: Training time for each model in various tasks (in hours).

| Model | `5m_vs_6m` (2M) | `1c3s5z` (2M) | `p5_vs_5` (5M) | `SurComb3` (5M) |
|-------|-----------------|---------------|----------------|-----------------|
| EMC   | 8.6  | 23.1 | 23.2  | 21.6* |
| MASER | 12.7 | 12.9 | 21.8  | 23.5  |
| RODE  | 6.0  | 10.5 | 15.0  | 20.6  |
| TRAMA | 9.1  | 10.5 | 12.8* | 15.1  |

## C    IMPLEMENTATION DETAILS

In this section, we present the implementation details of TRAMA. In TRAMA, we additionally consider the trajectory class $k$ as an additional condition along with timestep $t$ for selecting indices of quantized vectors during VQ-VAE training. We denote this indexing function as $\mathcal{J}(t, k)$. Algorithm 1 presents details of $\mathcal{J}(t, k)$.

---

**Algorithm 1** Compute $\mathcal{J}(t, k)$

---

1: **Input:** For given the number of codebook $n_c$, the maximum batch time $T$, the current timestep $t$, the number of trajectory class $n_{cl}$, and the index of trajectory class $k$
2: **if** $t == 0$ **then**
3:     $n_K = \lfloor n_c/n_{cl} \rfloor$
4:     $d = n_K/T$
5:     Keep the values of $n_K, d$ until the end of the episode
6: **end if**
7: $i_s = n_K \times (k - 1)$
8: **if** $d \geq 1$ **then**
9:     $\mathcal{J}(t, k) = i_s + \lfloor d \times t \rfloor : 1 : i_s + \lfloor d \times (t + 1) \rfloor$
10: **else**
11:     $\mathcal{J}(t, k) = i_s + \lfloor d \times t \rfloor$
12: **end if**
13: Return $\mathcal{J}(t, k)$

---

The computed $\mathcal{J}(t, k)$ for a given $(t, k)$ pair is then used for coverage loss in Eq. (5) to spread the quantized vectors throughout the embedding space of feasible states, $\chi$. Since Eq. (5) is expressed for a given $s_t^k$, we further elaborate on the expression considering batch samples. Modified VQ-VAE loss objective for given state $s_t$, nearest vector $x_{t,q} = [f_\phi^e(s_t)]_q$ and given class $k$ is expressed as follows.

$$\mathcal{L}_{VQ}^{tot}(\phi, \boldsymbol{e}, s_t, k) = \mathcal{L}_{VQ}(\phi, \boldsymbol{e}, s_t) + \lambda_{\mathrm{cvr}} \frac{1}{|\mathcal{J}(t,k)|} \sum_{j \in \mathcal{J}(t,k)} ||\mathrm{sg}[f_\phi^e(s_t)] - e_j||_2^2 \tag{12}$$

$$\begin{aligned}
\mathcal{L}_{VQ}(\phi, \boldsymbol{e}, s_t) = \\
||f_\phi^d([f_\phi^e(s_t)]_q) - s_t||_2^2 + \lambda_{\mathrm{vq}}||\mathrm{sg}[f_\phi^e(s_t)] - x_{t,q}||_2^2 + \lambda_{\mathrm{commit}}||f_\phi^e(s_t) - \mathrm{sg}[x_{t,q}]||_2^2
\end{aligned} \tag{13}$$

For batch-wise training for VQ-VAE, we train VQ-VAE with the following learning objective:

$$\mathcal{L}_{VQ}^{batch}(\phi, \boldsymbol{e}) = \frac{1}{B} \sum_{b=1}^{B} \sum_{t=0}^{T-1} \mathcal{L}_{VQ}^{tot}(\phi, \boldsymbol{e}, s_{t,b}, \bar{k}_b) \tag{14}$$

Algorithm 2 presents the overall training algorithm for all learning components of TRAMA, including $f_\phi^e$, $f_\phi^d$, $e$ in VQ-VAE; trajectory classifier $f_\psi$; trajectory-class predictor $\pi_\zeta$; trajectory-class representation model $f_\theta^g$; action policy $Q_\theta^i$ for $i \in I$; and $Q_\theta^{tot}$ for mixer network.

---

**Algorithm 2** Training algorithm for TRAMA

---

1: **Parameter:** Batch size $B$ for MARL training, batch size $M$ for classifier training, classifier update interval $n_\psi$, VQ-VAE update interval $n_\phi$, and the maximum training time $T_{env}$
2: **Input:** Individual Q-network $Q_\theta^i$ for $n$ agents, trajectory-class representation model $f_\theta^g$, VQ-VAE encoder $f_\phi^e$, VQ-VAE decoder $f_\phi^d$, VQ-VAE codebook $e$, trajectory-class predictor $\pi_\zeta$, replay buffer $D$, trajectory classifier $f_\psi$
3: Initialize network parameter $\theta, \phi, \psi, e$
4: $t_{env} = 0$
5: $n_{episode} = 0$
6: **while** $t_{env} \le T_{env}$ **do**
7:    Interact with the environment via $\epsilon$-greedy policy with $[Q_\theta^i]_{i=1}^n$ and get a trajectory $\tau_{s_{t=0}}$
8:    $t_{env} = t_{env} + t_{episode}$
9:    $n_{episode} = n_{episode} + 1$
10:    Encode $\tau_{s_{t=0}}$ by $f_\phi^e$ and get a quantized latent sequence $\tau_{\chi_{t=0}} = [f_\phi^e(\tau_{s_t})]_q$ by Eq. (1)
11:    Get indices sequence $\tau_{\mathcal{Z}_{t=0}}$ from $\tau_{\chi_{t=0}}$
12:    Append $\{\tau_{s_{t=0}}, \tau_{\mathcal{Z}_{t=0}}\}$ to $\mathcal{D}$
13:    Get $B$ sample trajectories $[\{\tau_{s_{t=0}}, \tau_{\mathcal{Z}_{t=0}}, \bar{k}\}]_{b=1}^B \sim \mathcal{D}$
14:    **for** $b \le B$ **do**
15:        **if** $\bar{\bar{k}}_b$ is None **then**
16:            Get trajectory-class label $\bar{k}_b$ via $f_\psi$
17:        **end if**
18:    **end for**
19:    **if** $\mathrm{mod}(n_{episode}, n_\phi)$ **then**
20:        Compute Loss $\mathcal{L}_{VQ}^{batch}(\phi, e)$ by Eq. (14) with $[\{\tau_{s_{t=0}}, \tau_{\mathcal{Z}_{t=0}}, \bar{k}\}]_{b=1}^B$
21:        Update $\phi, e$
22:    **end if**
23:    Compute Loss $\mathcal{L}(\theta, \zeta)$ by Eq. (10) with $[\{\tau_{s_{t=0}}, \tau_{\mathcal{Z}_{t=0}}, \bar{k}\}]_{b=1}^B$
24:    Update $\theta, \zeta$
25:    **if** $\mathrm{mod}(n_{episode}, n_\psi)$ **then**
26:        Get $M$ sample trajectories $[\tau_{\mathcal{Z}_{t=0}}]_{m=1}^M \sim \mathcal{D}$
27:        Compute a trajectory embedding $[\bar{e}^m]_{m=1}^M$
28:        Get class labels $\bar{K}$ by K-means clustering of $[\bar{e}^m]_{m=1}^M$
29:        Compute Loss $\mathcal{L}(\psi)$ by Eq. (7) with $[\bar{e}^m]_{m=1}^M$ and $\bar{K}$
30:        Update $\psi$
31:    **end if**
32: **end while**

---

## D ADDITIONAL EXPERIMENTS

### D.1 OMITTED EXPERIMENT RESULTS

In Section 5, we evaluate the various methods based on their mean return values instead of the mean win-rate. In single-task settings, both win-rate and return values show similar trends since agents learn policy to defeat enemies with a fixed unit combination under marginally perturbed initial positions. On the other hand, in multi-task settings, agents' policies may converge to suboptimality specialized on specific tasks, preventing them from gaining rewards in other tasks. In such a case, a high win-rate derived by specializing in certain tasks does not guarantee the generalization of policies in multi-task problems. Thus, we measure the mean return values instead. In the following, we present the omitted win-rate performance of experiments presented in Section 5.

In Figures 15 and 16, TRAMA shows the best or comparable performance compared to other baseline methods. However, the performance gap between TRAMA and other methods is not distinctively observed in these win-rate curves due to the aforementioned reason. Thus, measuring the mean return value is more suitable for multi-task problems.

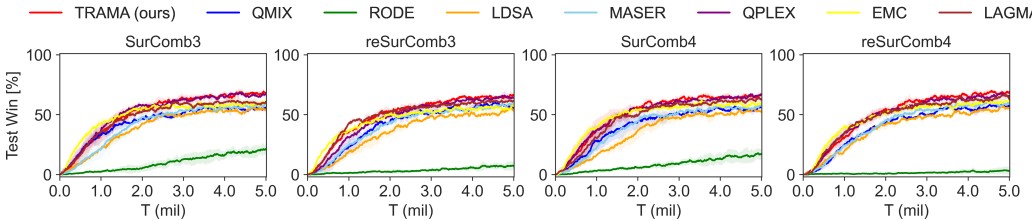

Figure 15: The mean win-rate of TRAMA compared to baseline algorithms on four multi-task problems presented in Table 1.

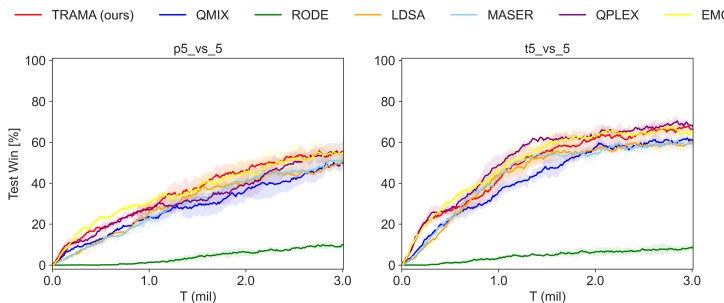

Figure 16: Performance comparison of TRAMA with win-rate against baseline algorithms on p5_vs_5 and t5_vs_5 in SMACv2. Here, $n_{cl}$=8 is assumed.

On the other hand, in Figure 17, similar performance trends are observed, illustrating the better performance of TRAMA in some tasks.

### D.2 PERFORMANCE COMPARISON WITH ADDITIONAL BASELINE METHODS

In this section, we present additional performance comparison with some omitted baseline methods, such as Updet (Hu et al., 2021) and RiskQ (Shen et al., 2023). Updet utilizes transformer architecture for agent policy based on the entity-wise input structure. Please note that this input structure is different from the conventional input settings of a single feature vector, which are widely adopted in MARL baseline methods (Rashid et al., 2018; Wang et al., 2020a; Zheng et al., 2021; Na & Moon, 2024). Therefore, we modified the input structure provided by the environment to evaluate Updet in SMACv2 tasks. RiskQ introduces the Risk-sensitive Individual-Global-Max (RIGM) to consider the common risk metrics such as the Value at Risk (VaR) metric or distorted risk measurements.

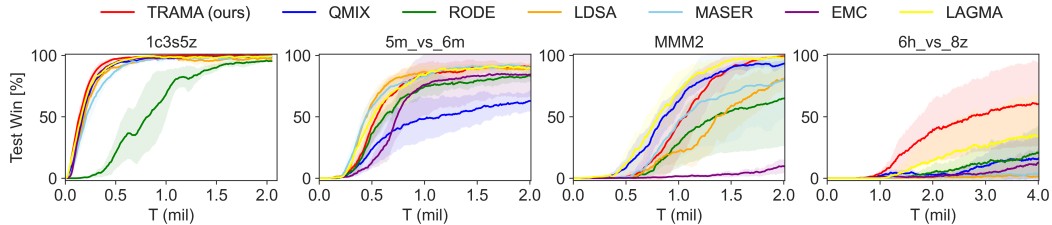

Figure 17: The mean win-rate of TRAMA compared to baseline algorithms on SMAC task.

For evaluation, we consider four multi-task problems presented in Table 1 and the conventional SMACv2 tasks, such as p5_vs_5 and t5_vs_5. We use the default settings presented in their codes for evaluation.

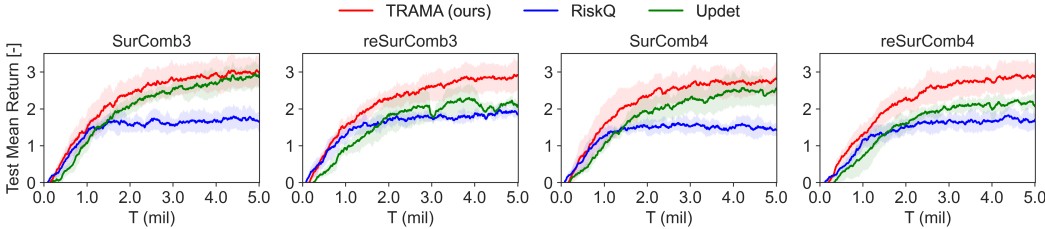

Figure 18: The mean return comparison of various models multi-task problems.

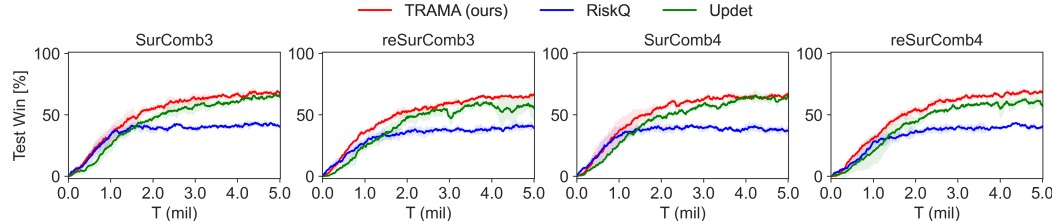

Figure 19: The mean win-rate comparison of various models on multi-task problems.

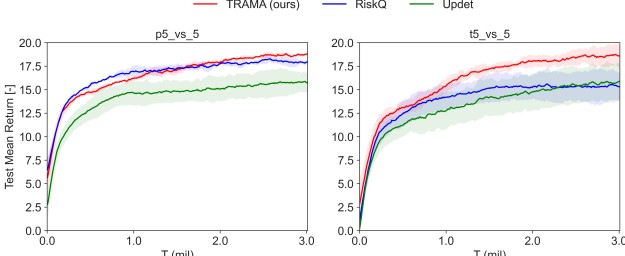

Figure 20: The mean return comparison of various models on SMACv2 p5_vs_5 and t5_vs_5.

In Figures 18 - 21, TRAMA shows the best performance compared to additional baseline methods, in terms of both return and win-rate in all multi-task problems.

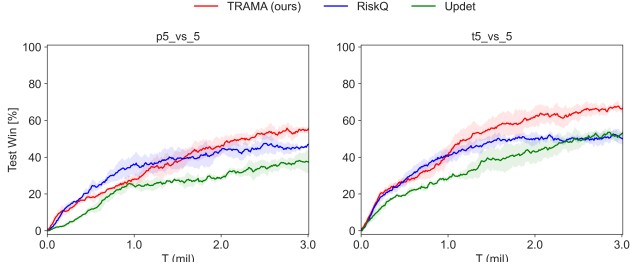

Figure 21: The mean win-rate comparison of various models on SMACv2 `p5_vs_5` and `t5_vs_5`.

### D.3 ADDITIONAL ABLATION STUDY

In this subsection, we present additional ablation studies on multi-task problems, such as `SurComb3` and `reSurComb4` to evaluate the impact of trajectory-class representation $g$ generated by $f_\theta^g$. To this end, we ablate $f_\theta^g$ in TRAMA and consider the one-hot vector, instead of $g^i$, as an additional condition to individual policies. We denote this model as **TRAMA (one-hot)**. In addition, we also ablate $J(t, k)$ and consider $J(t)$ to understand further the role of $J(t, k)$. Figure 22 illustrates the results.

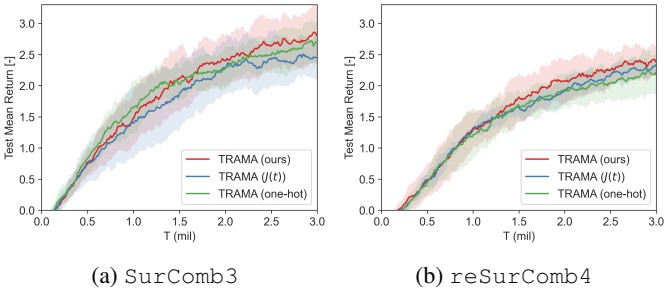

(a) `SurComb3`  (b) `reSurComb4`

Figure 22: Ablation studies on `SurComb3` and `reSurComb4`.

Similar to the results in Figure 12, $J(t, k)$ improves the performance as the quantized vectors are evenly distributed throughout $\chi$, yielding the clusters of trajectories with task similarities. On the other hand, the one-hot vector also gives additional information to the policy network as agents predict the trajectory class labels accurately, generating consistent signals to the policy for a given trajectory class. However, trajectory-class representation signifies this impact and further improves the performance of TRAMA.

We also conduct an additional ablation study on the conventional SMACv2 tasks (Ellis et al., 2024) to see the effectiveness of the trajectory-class representation. Figure 23 illustrates the result.

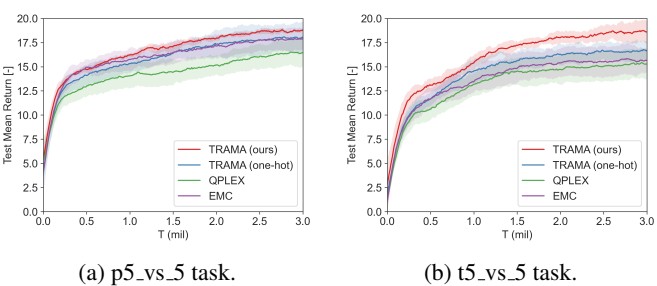

(a) p5_vs_5 task.  (b) t5_vs_5 task.

Figure 23: Additional ablation tests on trajectory-class representation, $g$.

In Figure 23, we can see that one-hot vector as an additional conditional information for decision-making also benefits the general performance. However, the trajectory-class representation signifies this gain, illustrated by the comparison between TRAMA (our) and TRAMA (one-hot).

## D.4 ADDITIONAL ABLATION STUDY ON CLUSTERING MODULE

In this section, we study the effect of the distribution of quantized vector throughout the embedding space, $\chi = \{x \in \mathbb{R}^d : x = f_\phi^e(s), s \in \mathcal{D}\}$. We utilize VQ-VAE embeddings to generate trajectory embeddings via Eq. 6, so that it can capture the commonality among trajectories. Although we do not have prior knowledge of $|\mathcal{K}|$, we can identify some tasks sharing similarity based on trajectory embeddings by assuming $n_{cl}$. In addition, we can utilize some adaptive algorithm to determine optimized $n_{cl}$. Please see Appendix D.5 for an adaptive clustering method.

In addition, even though $k_1$ and $k_2$ are actually different tasks if their differences are marginal, then they can be clustered into the same trajectory class. This mechanism is important since similar tasks may require a similar joint policy. Thus, having the exact trajectory class representation as an additional condition can be more beneficial in decision-making than having a vastly different trajectory-class representation, which could encourage different policies.

If quantized embedding vectors are not evenly distributed through the embedding space, the semantically dissimilar trajectories may share the same quantized vectors, which is unwanted. Without well-distributed quantized vectors, it becomes hard to construct distinct and meaningful clustering results. To see the importance of the distribution of embedding vectors in VQ-VAE, we ablate the coverage loss: (1) training with $\lambda_{cvr}$ considering $\mathcal{J}(t)$ only, and (2) TRAMA model, i.e., training with $\lambda_{cvr}$ considering $\mathcal{J}(t, k)$. In addition to Figure 4 in Section 3.1, Figure 24 presents additional ablation results on SMACv2 task *p5_vs_5*.

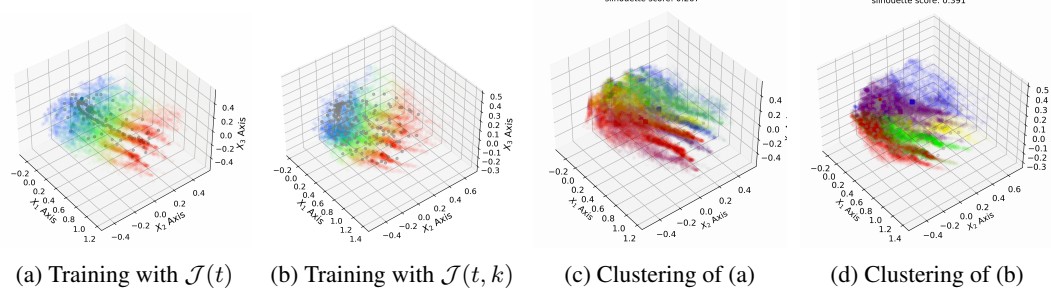

| (a) Training with $\mathcal{J}(t)$ | (b) Training with $\mathcal{J}(t, k)$ | (c) Clustering of (a) | (d) Clustering of (b) |

Figure 24: PCA of sampled trajectory embedding and VQ-VAE embedding vectors (gray circles). Colors from red to purple (rainbow) represent early to late timestep in (a) and (b). In (c) and (d), five clusters are assumed ($n_{cl} = 5$), and each trajectory embedding is colored with a designated class (red, green, blue, purple and yellow). `p5_vs_5` task is used for testing, and we ablate components related to coverage loss.

From Figures 4 and 24, we can see that having evenly distributed VQ-VAE embedding vectors is critical in clustering, which is highlighted by Silhouette score and the result of visualization. Since tasks in `p5_vs_5` can have marginal differences, the clustering results are unclear compared to the results from the customized multi-task problems in Figure 4.

## D.5 IMPLEMENTATION DETAILS OF CLUSTERING AND ADAPTIVE CLUSTERING METHOD

This section presents some details of the K-means++ clustering we adopt and a possible adaptive clustering method based on the Silhouette score. For centroid initialization, centroids are initially selected randomly from the data points. Then next centroid is selected probabilistically, where the probability of selecting a point is proportional to the square of its distance from the nearest existing centroid. At first, we iterate K-means with 10 different initial centroids. However, as we discussed in Section 3.2, generating coherent labels is also important. Thus, once we get the previous centroid value, we use this prior value as an initial guess for the centroid and run K-means just once with it.

As discussed in Section 5.3, we may select $n_{cl}$ large enough so that TRAMA can capture the possible diversity of unit combinations in multi-task problems. To consider some automatic update for $n_{cl}$,

we implement a possible adaptive clustering algorithm by adjusting $n_{cl}$ based on the Silhouette scores of candidate $n_{cl}$ values. We test this on multi-task problems and Figure 25 presents the result. We maintain other components in TRAMA the same. We initially assume $n_{cl} = 2$ and $n_{cl} = 6$ for adaptive methods. In Figure 25, the adaptive clustering method succeeded in finding optimal $n_{cl} = 3$ or $n_{cl} = 4$, in terms of Silhouette score. When an initial value $n_{cl} = 2$ is close to the optimal value, the overall performance in terms of return shows better performance as it quickly converges to the value $n_{cl} = 3$ and generates coherent label information for trajectory-class predictor.

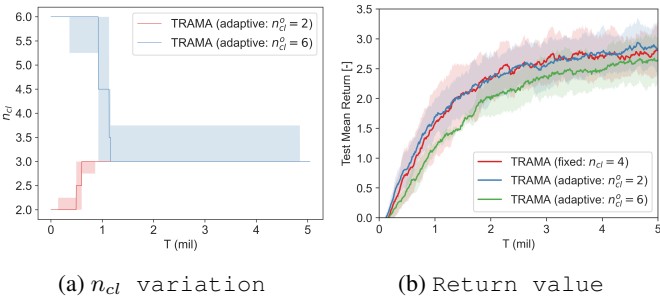

(a) $n_{cl}$ variation            (b) Return value

Figure 25: Test of adaptive clustering method on `SurComb4` multi-task problem.

Interestingly, the presented adaptive method converges to $n_{cl} = 3$ instead of $n_{cl} = 4$. In Figure 13, the clusters for `1c2s2z` and `3s2z` are close and share common units, such as `2s2z`. Thus, in terms of Silhouette score, $n_{cl} = 3$ has a marginally higher score and thus adaptive method, which converged to $n_{cl} = 3$, yields a similar performance in the case of fixed $n_{cl} = 4$.

D.6 EXPERIMENTS ON OUT-OF-DISTRIBUTION (OOD) TASKS

In this section, we present additional experiments on Out-Of-Distribution (OOD) tasks. Here, for $k_o \in \mathcal{K}_{ood}$, we define OOD tasks as $\mathcal{T}_{k_o}$ such that $\mathcal{T}_{k_o} \notin \mathcal{T}_{\text{train}}$ according to Definition 2.1. In other words, $\forall k_o \in \mathcal{K}_{ood}$ and $\forall k_i \in \mathcal{K}_{\text{train}}$, $S_{k_o} \cap S_{k_i}^c \neq \emptyset$ and $\mathbf{\Omega}_{k_o} \cap \mathbf{\Omega}_{k_i}^c \neq \emptyset$ should be satisfied. Thus, we can view *OOD tasks as unseen tasks*.

For this test, we use four models trained under different seeds for each method, and the corresponding win-rate and return curves are presented in Figure 26.

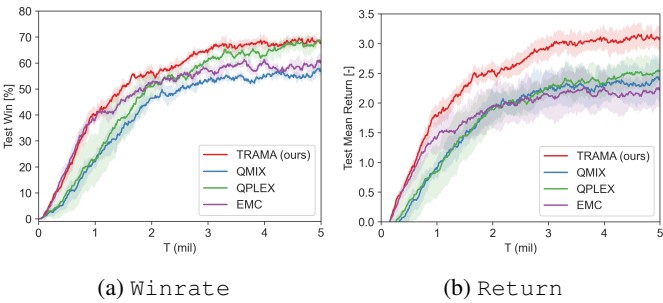

(a) Winrate  (b) Return

Figure 26: Performance comparison of various models on SurComb4.

In this test, we construct OOD tasks by differentiating either the unit combination or initial position distribution so that agents experience different observation distributions throughout the episode. Table 7 presents details of OOD tasks. We set two different types of unit combinations in OOD tasks, in addition to in-distribution (ID) unit combinations, (1) one similar to ID unit distribution and (2) the other largely different unit combinations. For example, unit combinations in OOD (#1) or OOD (#4) share some units with ID unit distribution. Specifically, 1c2s1z are common units in both the ID task with 1c2s2z and the OOD task with 1c3s1z. On the other hand, OOD (#2) or OOD (#5) accompanies largely different unit combinations.

In addition, we also construct OOD via largely different initial positions in OOD (#3) $\sim$ OOD (#5). Thus, we can expect that OOD (#5) is the most out-of-distributed task among various task settings in Table 7.

Table 7: Task configuration of OOD tests

| Name | Initial Position Type | Unit Combinations ($n_{comb}$) |
|---|---|---|
| ID (SurComb4) | Surrounded | $\{$1c2s2z, 3s2z, 2c3z, 2c3s$\}$ |
| OOD (#1) | Surrounded | $\{$1c4s, 1c3s1z, 2s3z$\}$ |
| OOD (#2) | Surrounded | $\{$5s, 5z, 5c$\}$ |
| OOD (#3) | Surrounded and Reflected | $\{$1c2s2z, 3s2z, 2c3z, 2c3s$\}$ |
| OOD (#4) | Surrounded and Reflected | $\{$1c4s, 1c3s1z, 2s3z$\}$ |
| OOD (#5) | Surrounded and Reflected | $\{$5s, 5z, 5c$\}$ |

Based on four models of each method trained with different seeds, we evaluate them across six different task settings, as shown in Table 7. For the evaluation, we run 50 test episodes per each trained model, resulting in a total of $4 \times 50 = 200$ runs per method. Tables 9 and 8 present test results. In the tables, the star marker (*) represents the best performance for a given task among various methods.

In Tables 9 and 8, TRAMA shows the better or comparable performance, in terms of both win-rate and return, in all cases including OOD tasks. In the OOD (#2) task, the differences in the win-rate are not evident compared to other baseline methods. This is reasonable because TRAMA cannot gain significant benefits from identifying similar task classes and encouraging similar policies through trajectory-class representations when there is no clear task similarity. In addition, TRAMA also shows the best performance in OOD tasks with highly perturbed initial positions, as in OOD (#3) $\sim$ OOD (#5).

Table 8: Return of OOD tests

|  | TRAMA | QMIX | QPLEX | EMC |
|---|---|---|---|---|
| ID (SurComb4) | **3.063**\*±0.489 | 2.201±0.398 | 2.323±0.119 | 2.348±0.406 |
| OOD (#1) | **2.706**\*±0.461 | 2.077±0.422 | 2.248±0.471 | 1.661±0.220 |
| OOD (#2) | **1.259**\*±0.148 | 0.939±0.463 | 1.049±0.240 | 0.774±0.326 |
| OOD (#3) | **2.307**\*±0.488 | 1.899±0.299 | 2.025±0.221 | 2.153±0.557 |
| OOD (#4) | **2.317**\*±0.280 | 1.484±0.219 | 1.993±0.199 | 1.457±0.127 |
| OOD (#5) | **0.921**\*±0.679 | 0.532±0.200 | 0.792±0.282 | 0.605±0.295 |

Table 9: Win-rate of OOD tests

|  | TRAMA | QMIX | QPLEX | EMC |
|---|---|---|---|---|
| ID (SurComb4) | **0.707**\*±0.025 | 0.540±0.059 | 0.667±0.034 | 0.613±0.025 |
| OOD (#1) | **0.625**\*±0.057 | 0.475±0.062 | 0.525±0.073 | 0.430±0.057 |
| OOD (#2) | **0.280**\*±0.020 | 0.270±0.057 | 0.265±0.017 | 0.275±0.057 |
| OOD (#3) | **0.620**\*±0.111 | 0.465±0.050 | 0.545±0.050 | 0.525±0.084 |
| OOD (#4) | **0.545**\*±0.052 | 0.355±0.071 | 0.475±0.059 | 0.365±0.033 |
| OOD (#5) | **0.220**\*±0.141 | 0.160±0.037 | 0.215±0.050 | 0.205±0.062 |

**Discrepancy between return and win-rate in multi-task problems** In Tables 8 and 9, the win-rate difference (in ratio) between TRAMA and QPLEX is about 6% while the return difference (in ratio) is about 32%.

This discrepancy can be observed in multi-task problems since a trained model can be specialized on some tasks yet less effective in other tasks, making not much reward. For example, both models A and B succeed in solving task 1 while failing on task 2 at the same frequency. However, if model A nearly succeeds in solving task 2 but ultimately fails, while model B completely fails in solving task 2, this scenario results in a similar win-rate for A and B but a different return.

In another case, both models A and B become specialized in one of the tasks, but the total return from tasks 1 and 2 can differ. If task 1 yields a larger return compared to task 2 for success, and model A overfits to task 1 while model B overfits to task 2, this scenario results in a similar win rate for A and B but a different return, i.e., a larger return for model A.

In both cases, model A is preferred over model B. This highlights why it is important to focus on the return difference when evaluating model performance in multi-task problems.

**Qualitative analysis on trajectory class prediction in OOD tasks** To understand how TRAMA predicts trajectory classes in OOD tasks, we first evaluate how accurately agents in TRAMA predict trajectory classes in in-distribution (ID) tasks.

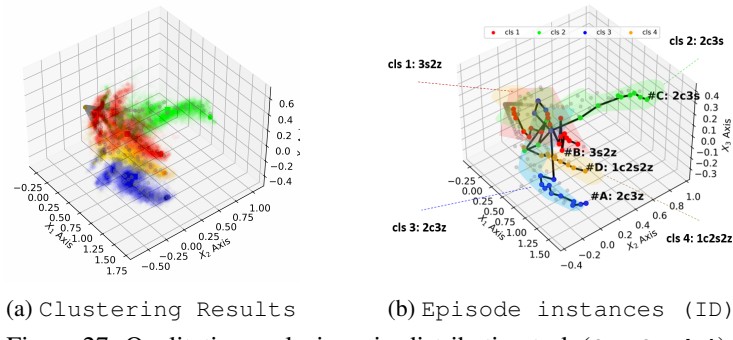

(a) Clustering Results   (b) Episode instances (ID)

Figure 27: Qualitative analysis on in-distribution task (SurComb4).

Figure 27(a) presents the clustering results of the ID task to determine which cluster corresponds to which type of task. From this result, we identify types of task denoted by each trajectory class

in Figure 27(b) and present some test episodes #A ~ #D. We also present the overall prediction made by agents across all timesteps in each test episode by the conventional box plot. The box plot denotes 1st (Q1) and 3rd (Q3) quartiles with a color box and median value with a yellow line within the color box.

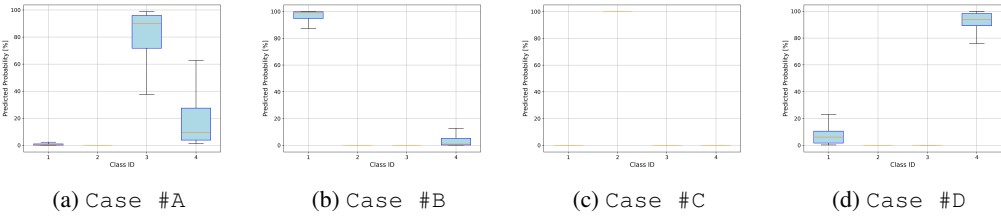

(a) Case #A     (b) Case #B     (c) Case #C     (d) Case #D

Figure 28: Overall prediction on trajectory class made by agents (ID task). Please refer to Figure 27 for each episode case.

In Figure 28, agents are confident in predicting trajectory class. In Case #A, agents predict a possibility that the given task belongs to class 4 instead of class 3, as their unit combinations can be quite similar, such as `1c2z`, when some units are lost.

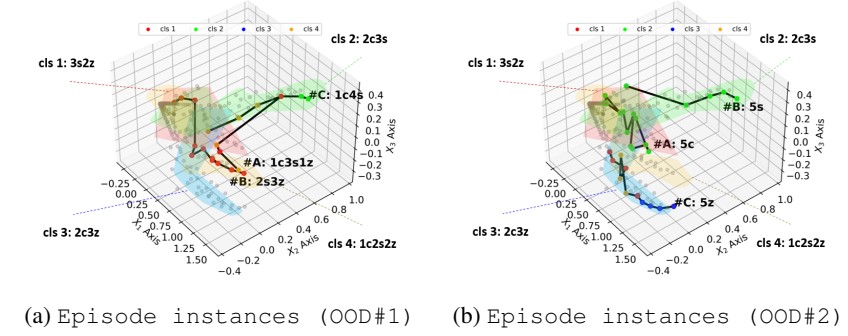

(a) Episode instances (OOD#1)     (b) Episode instances (OOD#2)

Figure 29: Qualitative analysis on out-of-distribution tasks (OOD #1 and OOD #2).

Figures 29 present qualitative analysis results of out-of-distribution tasks, OOD#1 and OOD#2 and Figures 30 and 31 illustrate the predictions made by agents for each test case. In Figure 30, agents predict the class of a given out-of-distribution task as the closest class experienced during training. Some tasks in OOD#1 share some portion of unit combinations, yielding strong predictions on Case #B and Case #C. This may yield *OOD task adaptation*, as a predicted trajectory class representation can encourage a joint policy that is effective in tasks sharing some units with ID tasks.

On the other hand, when there is no evident similarity between OOD tasks and ID tasks, agents make weak predictions on trajectory class as presented in Figure 31(c). In this case, we can *detect highly OOD tasks* by setting a certain threshold of confidence level, such as 50%.

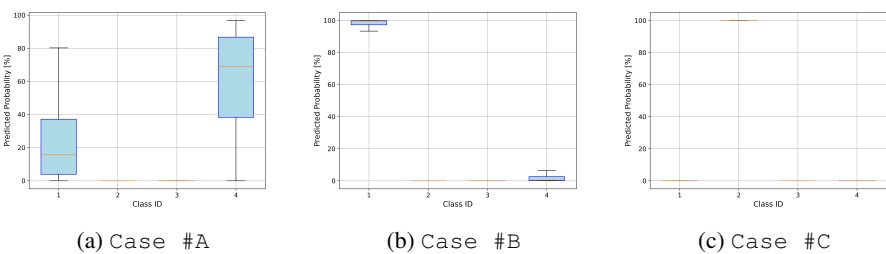

(a) Case #A     (b) Case #B     (c) Case #C

Figure 30: Overall prediction on trajectory class made by agents (OOD#1 task). Please refer to Figure 29(a) for each episode case.

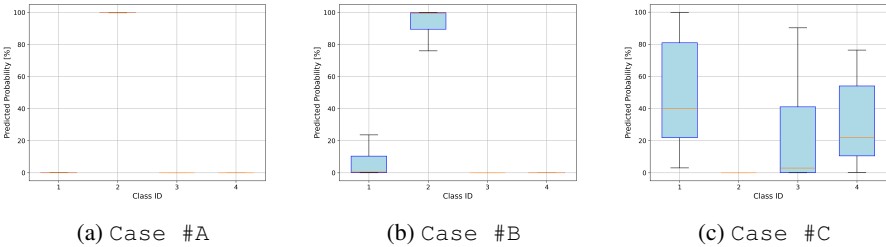

Figure 31: Overall prediction on trajectory class made by agents (OOD#2 task). Please refer to Figure 29(b) for each episode case.

## D.7 TRAJECTORY CLASS PREDICTION

In this section, we present the omitted results of trajectory class predictions made by agents. We present the accuracy of predictions in ratio. Tables 10, 11, and 12 demonstrate the prediction accuracy of each training time ($t_{env}$). Columns of each Table represent each timestep ($t_{episode}$) within episodes. Throughout various multi-task problems, agents accurately predict the trajectory class. In the results, agents predict more accurately as the timestep proceeds since they get more information for prediction through their observations.

Table 10: The accuracy of trajectory class prediction (1M)

| $t_{env}$=1M | $t_{episode}$=0 | $t_{episode}$=10 | $t_{episode}$=20 | $t_{episode}$=30 |
|---|---|---|---|---|
| SurComb3 | 0.841±0.062 | 0.855±0.044 | 0.861±0.044 | 0.886±0.043 |
| SurComb4 | 0.766±0.085 | 0.791±0.104 | 0.848±0.096 | 0.876±0.088 |
| reSurComb3 | 0.980±0.014 | 0.991±0.011 | 0.997±0.004 | 0.997±0.004 |
| reSurComb4 | 0.838±0.092 | 0.900±0.044 | 0.920±0.055 | 0.934±0.046 |

Table 11: The accuracy of trajectory class prediction (3M)

| $t_{env}$=3M | $t_{episode}$=0 | $t_{episode}$=10 | $t_{episode}$=20 | $t_{episode}$=30 |
|---|---|---|---|---|
| SurComb3 | 0.880±0.038 | 0.909±0.062 | 0.925±0.041 | 0.930±0.04 |
| SurComb4 | 0.908±0.063 | 0.939±0.025 | 0.941±0.023 | 0.947±0.022 |
| reSurComb3 | 0.947±0.046 | 0.961±0.029 | 0.975±0.017 | 0.968±0.023 |
| reSurComb4 | 0.889±0.062 | 0.927±0.037 | 0.933±0.045 | 0.931±0.042 |

Table 12: The accuracy of trajectory class prediction (5M)

| $t_{env}$=5M | $t_{episode}$=0 | $t_{episode}$=10 | $t_{episode}$=20 | $t_{episode}$=30 |
|---|---|---|---|---|
| SurComb3 | 0.853±0.034 | 0.902±0.046 | 0.933±0.052 | 0.945±0.044 |
| SurComb4 | 0.925±0.084 | 0.939±0.070 | 0.963±0.049 | 0.957±0.045 |
| reSurComb3 | 0.963±0.071 | 0.953±0.094 | 0.967±0.066 | 0.972±0.056 |
| reSurComb4 | 0.886±0.024 | 0.913±0.017 | 0.917±0.035 | 0.930±0.027 |

