# OpenReview forum: "Trajectory-Class-Aware Multi-Agent Reinforcement Learning"
_ICLR.cc/2025/Conference — ICLR 2025 Poster_

### Official Review · Reviewer_YpdJ · 2024-10-24

**Soundness:** 3
**Presentation:** 2
**Contribution:** 3
**Rating:** 6
**Confidence:** 4

**Summary:**

This paper studies multi-task multi-agent reinforcement learning. It proposes a trajectory clustering-based approach, where trajectories from different environments or tasks are clustered based on similarities in the latent space. The clustering label is then used as an additional input for the policy。

**Strengths:**

Multi-task multi-agent reinforcement learning is an important challenge for MARL. The proposed trajectory clustering approach is innovative and offers a novel solution.

**Weaknesses:**

[1] The primary concern is whether the clustering approach is truly necessary. From my understanding, the purpose of clustering is merely to provide a label as additional information for the policy. It seems that the proposed approach only evaluates tasks seen during training, without considering unseen tasks during test time. Does this approach generalize to unseen tasks? If so, could the authors provide experimental results to support this? If not, why is the clustering necessary to provide a label, since we could easily have labels for the tasks during training and use it to train the trajectory class predictor? Perhaps simply using the task ID as a one-hot vector could replace the entire clustering module, as this has been effective in some single-agent RL work, see 1'.

[2] The notations in Section 3 are unclear and could lead to confusion. Please consider improving them.

[3] I think some discussions and comparisons with Updet ,see 2', is necessary, since Updet could be thought of a multi-task marl algorithom



1' Hansen, N., Su, H., & Wang, X., 2023. TD-MPC2: Scalable, robust world models for continuous control. arXiv preprint arXiv:2310.16828

2' Hu, S., Zhu, F., Chang, X., & Liang, X. (2021). Updet: Universal multi-agent reinforcement learning via policy decoupling with transformers. arXiv preprint arXiv:2101.08001.

**Questions:**

[1] It seems that the policy module does not propagate gradients back to the clustering module. I wonder why these two modules are not coupled, allowing the clustering module to have some trainable components that could help extract useful information for the policy?

[2] The paper emphasizes the coverage loss significantly. However, why is distributing throughout the embedding space considered beneficial?

[3] In equation (7), why do you directly sum the embedding vectors? Does this make sense, or could there be better ways to handle this?

[4] How do you manage different observation and input dimensions at the same time for different tasks?

[5] "To evaluate performance, we consider the overall return value instead of the win-rate, as the learned policy may be specialized for specific goal tasks while being less effective for others in multi-goal tasks." Could you elaborate further on this point?

[6] For comparison with QMIX, do you use PyMARL2 (see 1') or PyMARL? PyMARL2 fine-tunes the parameters for QMIX and shows better performance. For example, in Figure 10, in the 6h_vs_8z scenario, based on my experience, QMIX could achieve a better return at 4M environment steps.


1', Hu, J., Jiang, S., Harding, S. A., Wu, H., & Liao, S. W. (2021). Rethinking the implementation tricks and monotonicity constraint in cooperative multi-agent reinforcement learning. arXiv preprint arXiv:2102.03479

---

> ### Author Response · Authors · 2024-11-24
> **Response #1**
>
> **W1. Regarding the necessity of clustering and further generalizability.**
>
>
> **A1.** Thanks for the comment.
>
> First, we want to clarify that there is **no distinct label provided by the environment** in MARL multi-task settings we are solving (please see Section 2.2 for the formal definition). This can be viewed as a more generalized setting than conventional multi-task settings, where task ID is often provided.
>
> Clustering module aims to identify meaningful trajectory classes based on trajectory embeddings. We do not know which type of class a given trajectory belongs to and even how many tasks exist. We just assume the number of classes and conduct clustering with this assumption. Then, TRAMA automatically figures out the similarities between trajectories and clusters them into certain classes.
>
> For example, In **Figure 13** of the revised or original manuscript, the environment does not provide trajectory class labels during training. Still, TRAMA identifies four branches that share the commonalities among trajectories within the same class. Notably, the initial guess of the number of classes was 8, but TRAMA effectively identifies there are 4 meaningful classes or tasks.
>
> In addition, unlike single-agent tasks, agents are only accessible to their partial observation in MARL settings. That is the main reason why "trajectory-class predictor" is introduced. Agents need to figure out which type of tasks they are experiencing in the execution phase based on the agent-wise predictor model.
>
> As the reviewer mentioned, coherent one-hot information can also benefit conditional policy. However, **Figures 12 and 23 of Appendix D.3** in the revised manuscript illustrate that adopting trajectory-class representation is more beneficial compared to the case utilizing simple one-hot as an additional condition.
>
> In addition, to further support the benefits of trajectory clustering and learning trajectory-class representation and the generalizability of TRAMA in unseen tasks, we conducted additional experiments on out-of-distribution tasks in **Appendix D.6** of the revised manuscript. Please refer to the results for details. The following tables summarize the OOD tests.
>
> # Task Configuration of OOD Tests
> | Name       | Initial Position Type    | Unit Combinations           |
> |------------|--------------------------|-----------------------------|
> | ID (SurComb4) | Surrounded              | {1c2s2z, 3s2z, 2c3z, 2c3s} |
> | OOD (#1)   | Surrounded               | {1c4s, 1c3s1z, 2s3z}        |
> | OOD (#2)   | Surrounded               | {5s, 5z, 5c}                |
> | OOD (#3)   | Surrounded and Reflected | {1c2s2z, 3s2z, 2c3z, 2c3s} |
> | OOD (#4)   | Surrounded and Reflected | {1c4s, 1c3s1z, 2s3z}        |
> | OOD (#5)   | Surrounded and Reflected | {5s, 5z, 5c}                |
>
> # Return of OOD Tests
> |            | TRAMA     | QMIX   | QPLEX  | EMC    |
> |------------|-----------|--------|--------|--------|
> | ID (SurComb4) | **3.063**$^*$ ± 0.489 | 2.201 ± 0.398 | 2.323 ± 0.119 | 2.348 ± 0.406 |
> | OOD (#1)   | **2.706**$^*$ ± 0.461 | 2.077 ± 0.422 | 2.248 ± 0.471 | 1.661 ± 0.220 |
> | OOD (#2)   | **1.259**$^*$ ± 0.148 | 0.939 ± 0.463 | 1.049 ± 0.240 | 0.774 ± 0.326 |
> | OOD (#3)   | **2.307**$^*$ ± 0.488 | 1.899 ± 0.299 | 2.025 ± 0.221 | 2.153 ± 0.557 |
> | OOD (#4)   | **2.317**$^*$ ± 0.280 | 1.484 ± 0.219 | 1.993 ± 0.199 | 1.457 ± 0.127 |
> | OOD (#5)   | **0.921**$^*$ ± 0.679 | 0.532 ± 0.200 | 0.792 ± 0.282 | 0.605 ± 0.295 |
>
>
> # Win-rate of OOD Tests
> |            | TRAMA     | QMIX   | QPLEX  | EMC    |
> |------------|-----------|--------|--------|--------|
> | ID (SurComb4) | **0.707**$^*$ ± 0.025 | 0.540 ± 0.059 | 0.667 ± 0.034 | 0.613 ± 0.025 |
> | OOD (#1)   | **0.625**$^*$ ± 0.057 | 0.475 ± 0.062 | 0.525 ± 0.073 | 0.430 ± 0.057 |
> | OOD (#2)   | **0.280**$^*$ ± 0.020 | 0.270 ± 0.057 | 0.265 ± 0.017 | 0.275 ± 0.057 |
> | OOD (#3)   | **0.620**$^*$ ± 0.111 | 0.465 ± 0.050 | 0.545 ± 0.050 | 0.525 ± 0.084 |
> | OOD (#4)   | **0.545**$^*$ ± 0.052 | 0.355 ± 0.071 | 0.475 ± 0.059 | 0.365 ± 0.033 |
> | OOD (#5)   | **0.220**$^*$ ± 0.141 | 0.160 ± 0.037 | 0.215 ± 0.050 | 0.205 ± 0.062 |
>
> Notably, TRAMA shows strong performance in OOD tasks containing some similarity (OOD\#1 and OOD\#4) with in-distribution tasks. We speculate the reason as TRAMA gains some benefits from identifying similar tasks from experience during training and encouraging similar policies through trajectory-class representation. In this way, TRAMA can adapt to OOD tasks.

---

> ### Author Response · Authors · 2024-11-24
> **Response #2**
>
> **W2. Regarding the notation.** The notations in Section 3 are unclear and could lead to confusion. Please consider improving them.
>
> **A2.** Thanks for the comment. We proofread the manuscript and revised the manuscript to make the contents more clear. For example, we replace $D$ with $d$ to denote dimension of embedding vectors to prevent confusion $D$ from $\mathcal{D}$, the replay buffer.
>
> Please let us know if there are still unclear parts.
>
> **W3. Regarding additional discussion.** I think some discussions and comparisons with Updet , see [2], is necessary, since Updet could be thought of a multi-task marl algorithom
>
> [2] Hu, S., Zhu, F., Chang, X., and Liang, X. (2021). Updet: Universal multi-agent reinforcement learning via policy decoupling with transformers. arXiv preprint arXiv:2101.08001.
>
> **A3.** Thanks for the comment. However, the listed work (Updet) requires the decoupling process by modifying the environment setting to make feature-based observation into entity-based observation. This could be useful in multi-task settings but due to the preprocessing and different environment settings, we deem this approach as a solution for different types of tasks, where entity-based information is viable. For that reasons, the mentioned work is not widely compared within standard cooperative MARL tasks with a single feature vector observation input. Please note that dividing a feature vector into entity-based inputs requires domain-specific knowledge and task environment modification.
>
> We present additional experiment to compare TRAMA with other baselines including Updet in **Appendix D.2** of the revised manuscript. TRAMA shows better performance compared to Updet in all test cases.
>
> **Q1. Regarding training method of clustering module.** It seems that the policy module does not propagate gradients back to the clustering module. I wonder why these two modules are not coupled, allowing the clustering module to have some trainable components that could help extract useful information for the policy?
>
> **A4.** The objectives and requirements of clustering module and policy learning are different. Clustering requires overall trajectories stored in the replay buffer $\mathcal{D}$ so that explored trajectories are well clustered. On the other hand, MARL training is conducted only with sampled batches size of 32. If we use all trajectories in $\mathcal{D}$ to training MARL and clustering at the same time is not practical for MARL training. Conversely, utilizing 32 batch samples for clustering yields too myopic clustering results. This is one of the main reasons why the two training processes must be decoupled.
>
> In addition, clustering results are required to train the agent-wise trajectory-class-predictor. Training both at the same time may cause instability. Moreover, the trainable component is contained in trajectory-class representation model. This model is directly trained with policy training, which could extract useful information for the policy, as the reviewer mentioned.
>
> **Q2. Regarding the significance of coverage loss when constructing embedding space.** The paper emphasizes the coverage loss significantly. However, why is distributing throughout the embedding space considered beneficial?
>
> **A5.** Thanks for the comment. Distribution of VQ-VAE embedding vectors evenly throughout the embedding space is important since it generates trajectory embedding.
>
> We want to utilize the commonality among trajectories via quantized embedding vectors effectively. If quantized embedding vectors are not evenly distributed through the embedding space, the semantically dissimilar trajectories may share the same quantized vectors, which is unwanted. Without well-distributed quantized vectors, it becomes hard to construct distinct and meaningful clustering results.
>
> To illustrate this, **Figure 4** in the revised manuscript and **Appendix D.4** present how clustering is influenced by the distribution of VQ-VAE embedding vectors.
>
> **Q3. Regarding trajectory embedding.** In equation (7), why do you directly sum the embedding vectors? Does this make sense, or could there be better ways to handle this?
>
> **A6.** This trajectory embedding via summation is one of simple forms. We could introduce learnable factors to consider different importance weights for states at each timestep. However, it may also introduce the additional complexity to the proposed framework. The main objective of this trajectory embedding is to generate distinct and meaningful clustering, ultimately to help the overall MARL training. Thus, we want to simplify the additional component as simple as possible. In addition, after some tests, we confirmed that this simple trajectory embedding is sufficient to generate effective clustering.

---

> ### Author Response · Authors · 2024-11-24
> **Response #3**
>
> **Q4. Regarding experiment details.** How do you manage different observation and input dimensions at the same time for different tasks?
>
> **A7.** Multi-task settings in MARL contain different unit combinations and highly different initial positions, unlike the original SMAC task. However, each task shares the input dimension while adopting different contents in observation.
>
> **Q5. Regarding clarification on explanation in the manuscript.** "To evaluate performance, we consider the overall return value instead of the win-rate, as the learned policy may be specialized for specific goal tasks while being less effective for others in multi-goal tasks." Could you elaborate further on this point?
>
> **A8.** To validate this, please see **Tables 8 and 9in Appendix D.6** with OOD tasks or  Return of OOD Tests and Win-rate of OOD Tests in Response #1 above. The win-rate difference (in ratio) between TRAMA and QPLEX is about 6\%, while the return difference
> (in ratio) is about 32\%.
>
> This discrepancy can be observed in multi-task problems since a trained model can be specialized on some tasks yet less effective in other tasks, making not much reward. For example, both models A and B succeed in solving task 1 while failing on task 2 at the same frequency. However, if model A nearly succeeds in solving task 2 but ultimately fails, while model B completely fails in solving task 2, this scenario results in a similar win-rate for A and B but a different return.
>
> In another case, both models A and B become specialized in one of the tasks, but the total return from tasks 1 and 2 can differ. If task 1 yields a larger return compared to task 2 for success, and model A overfits to task 1 while model B overfits to task 2, this scenario results in a similar win rate for A and B but a different return, i.e., a larger return for model A.
>
> In both cases, model A is preferred over model B. This highlights why it is important to focus on the return difference when evaluating model performance in multi-task problems.
>
> **Q6. Regarding one of baseline models.** For comparison with QMIX, do you use PyMARL2 ([1]) or PyMARL? PyMARL2 fine-tunes the parameters for QMIX and shows better performance. For example, in Figure 10, in the 6h\_vs\_8z scenario, based on my experience, QMIX could achieve a better return at 4M environment steps.
>
> [1] Hu, J., Jiang, S., Harding, S. A., Wu, H., and Liao, S. W. (2021). Rethinking the implementation tricks and monotonicity constraint in cooperative multi-agent reinforcement learning. arXiv preprint arXiv:2102.03479
>
> **A9.** Thanks for the comment. We used the configuration presented in the original QMIX paper, not [1]. In [1], two types of finetuning methods are introduced for 6h\_vs\_8z task: $Q(\lambda)$ and $\epsilon$-search. Since all baseline methods utilize one-step TD error for Q-learning, we present the corresponding results for fair comparison.
>
> For $\epsilon$ value, because each algorithm may have a different optimal $\epsilon$ for exploration, we finetuned the epsilon rate for QMIX as presented in [1] and updated the results accordingly. Please check updated **Figures 8 and 17** in the revised manuscript.

---

> > ### Comment · Reviewer_YpdJ · 2024-11-25
> >
> > I appreciate your efforts in responding to my questions. I still have some questions about the multi-task setting in your work.
> >
> > (1) For SMAC v2, during training, we could know the unit combinations when the game starts, is this right? If so, then we already have some information regarding which subtask we are solving, even though initial positions could vary. That said, maybe we could still have some form of subtask labels during training in SMAC v2?
> >
> > (2) In your definition, why do different subtasks share the same transition probability? I just want to double check, because I feel different unit combinations might have different underlying dynamics, so different transition probabilities?
> >
> > (3) I think the problem the authors are trying to tackle is a subset of general multi-task RL, is this right? Because different subtasks share the same transition probabilitiies. I wonder since the idea of trajectory clustering is pretty straightforward and should generalize to settings of tasks with different rewards and transition probabilities?  I have a feeling the current setting is somehow tailored for the SMAC v2 environment. No need to conduct any new experiments due to limited time, but some discussions might be interesting.

---

> > > ### Author Response · Authors · 2024-11-25
> > >
> > > Thank you for the additional comments and active participation. We really appreciate that.
> > >
> > > **Q1.** For SMAC v2, during training, we could know the unit combinations when the game starts, is this right? If so, then we already have some information regarding which subtask we are solving, even though initial positions could vary. That said, maybe we could still have some form of subtask labels during training in SMAC v2?
> > >
> > > **A1.** Yes, by extracting some relevant information from global state, we can identify the unit combination of a given task. However, this identification is not sufficient information to learn a generalized policy because such identification should be evaluated for similarities across unit combinations as well as initial positions. Eventually, unsupervised learning on such task similarities will be required, and our method responds to that necessity.
> > >
> > > Additionally, **When possible unit combinations increases, we cannot give such supervisions, the reviewer suggested.** For example, in t5\_vs\_5 task of SMACv2, there can be total $3^5=243$ unit combinations. In such cases, utilizing task label solely based on unit combination is limited since adopting $243$ conditional information for decision-making may yield instability and hardly make some effective coordination. On the other hand, TRAMA automatically clusters certain tasks, which share some similarities among them, into the same class and encourage same joint policy for these tasks, yielding better generalizable policy learning as illustrated by example in SMACv2 tasks, p5\_vs\_5 and t5\_vs\_5 (**Figure 7, Figure 20, Figure 21 in the revised manuscript**).
> > >
> > > In detail, there could be issues in generalization. For example, highly similar trajectories from different tasks, such as task\#1 and task\#2, are classified by either task\#1 or task\#2 via clustering or classifier of TRAMA. This can occur when two tasks share the majority of their unit combinations. In such cases, an agent-wise predictor also predicts trajectory-class-type (task\#1 or task\#2) with low confidence, resulting in similar joint policies for given observations or indifferent joint policy for the trajectory-class-representation of task\#1 and task\#2. This could represent an important generalization mechanism in TRAMA, as similar situations may benefit from similar joint policies rather than distinct ones. However, we cannot expect this when adopting hard labeling for task\#1 and task\#2. This aspect should also be evaluated for hard labeling method solely based on unit combinations.
> > >
> > > **Q2.** In your definition, why do different subtasks share the same transition probability? I just want to double check, because I feel different unit combinations might have different underlying dynamics, so different transition probabilities?
> > >
> > > **A2.** Thank you for discussing interesting point. The major difference between single-agent multi-task and multi-agent multi-task comes from partial observability. We denote different tasks share the transition and reward function as they are still in the same domain, which means that they share underlying dynamics of units. Within the same domain, we discard the case where the same unit, for example Zealot (with short attack range), suddenly make different action consequences (e.g. long-range attack). If the global state does not contain any unit type information, then state transition function may behave differently according to tasks or unit combination of the task, as the reviewer speculate. However, that is not our case.
> > >
> > > In the perspective of partial observation of each agent, transition and reward function obviously seems to change according to observation condition and types of tasks. One could simply define different transition function and reward function for each task, but a more strict definition is like Section 2.2 for multi-agent multi-task settings, such as SMACv2.

---

> ### Author Response · Authors · 2024-11-25
>
> **Q3.** I think the problem the authors are trying to tackle is a subset of general multi-task RL, is this right? Because different subtasks share the same transition probabilitiies. I wonder since the idea of trajectory clustering is pretty straightforward and should generalize to settings of tasks with different rewards and transition probabilities? I have a feeling the current setting is somehow tailored for the SMAC v2 environment. No need to conduct any new experiments due to limited time, but some discussions might be interesting.
>
> **A3.** Yes. Our setting would be classified as a special case of **multi-agent** multi-task RL (under decPOMDP), if multi-agent multi-task RL contain cases with task-dependent transition, reward, and observation functions.
>
> Multi-agent multi-task problems have not been not explored much through common complex benchmark problems, such as SMAC. We have searched for others providing multi-agent multi-task settings under decPOMDP settings but could not find any comparable benchmark problems, so far.
>
> There could be other way to test generalization in multi-agent multi-task by adopting task-specific $P_k$, $R_k$, $O_k$, leaving support state space $S$ not quite change, as in single-agent multi-task settings. On the other hand, in our problem, we explore the task dependent $S_k$, which yields task specific joint observation space ${\Omega_k}$. This could generate significant variation in observations, making agents hard to learn generalizable policies. Please note that decPOMDP already contains stochasticity in agent's transition due to stochastic actions of agents, and thus learning generalizable policy in multi-tasks, sharing underlying transition and reward function, is still challenging. TRAMA effectively deals with this kind of problems.
>
> (Regarding clustering) In addition, there could be other way of trajectory clustering. However, to effectively capture certain similarities between trajectories, neglecting some marginal differences, we adopt VQ-VAE for quantized latent space construction. To this end, we introduce some modification via modified coverage loss considering trajectory class, for better quality of trajectory embeddings. (**Figure 4 and Appendix D.4**). This module is important in cluster similar trajectories into the same class. We hope the reviewer understand this point, too.

---

> > ### Comment · Reviewer_YpdJ · 2024-11-25
> >
> > Related to Q1: For SMAC v2, during training, we could know the unit combinations when the game starts, correct? If so, then we already have some information regarding which subtask we are solving, even though initial positions may vary. That said, perhaps we could still have some form of subtask labels during training in SMAC v2?
> >
> > I see the advantages of your approach. I am just suggesting that integrating hard task labels into your clustering framework could potentially help if you can obtain those task IDs.
> >
> > Related to Q2: In your definition, why do different subtasks share the same transition probability? I just want to confirm because I feel different unit combinations might have different underlying dynamics, and thus different transition probabilities.
> >
> > I see that you include unit types as a component of your state, so your definition works fine—indeed, all subtasks share the same transition probabilities in this case.
> >
> > Overall, I appreciate the authors' efforts and think there is certainly value in this work. I hope to see future work on this topic, perhaps in a more generalized multi-agent multi-task setting. Hence, this work is above my personal acceptance threshold. I would increase my score to 6.

---

> > > ### Author Response · Authors · 2024-11-26
> > >
> > > We sincerely appreciate your efforts in reviewing the paper and contributing to the discussions. Your constructive comments have greatly helped us enhance its quality and clarity. Thank you once again.

---

### Official Review · Reviewer_q68e · 2024-11-04

**Soundness:** 3
**Presentation:** 3
**Contribution:** 3
**Rating:** 6
**Confidence:** 4

**Summary:**

This paper proposes a novel multi-agent reinforcement learning framework called TRAMA (TRajectory-class-Aware Multi-Agent reinforcement learning) to address the issue of policy generalization in multi-goal tasks. TRAMA introduces the idea of trajectory clustering, where trajectories are embedded and clustered to enable agents to recognize the type of task they are experiencing based on their trajectories, and use this prediction as additional information for decision-making, thereby learning policies that can adapt to different goal tasks. The main components of TRAMA include: 1) constructing a quantized latent space using a modified VQ-VAE to generate trajectory embeddings; 2) performing trajectory clustering based on the embeddings; 3) training a trajectory-class predictor and class representation model to generate trajectory-class-aware policies. TRAMA achieves better performance than multiple baseline methods on various multi-goal tasks in StarCraft II.

**Strengths:**

1. TRAMA effectively addresses the policy generalization problem in multi-goal tasks. Traditional MARL methods often learn policies that are only effective for specific tasks and lack generalizability. By enabling agents to predict trajectory classes and learn class-related policy representations, TRAMA significantly improves the adaptability of policies, which is validated by the experimental results showing that TRAMA achieves higher average returns than other methods on multiple multi-goal tasks (Figure 6).
2. The modified VQ-VAE quantization method better learns trajectory embeddings. By considering both time steps and trajectory classes in the coverage loss, the quantized vectors are more evenly distributed in the embedding space of different trajectory classes (Figure 3), providing a good foundation for subsequent clustering.
3. Trajectory clustering and the class predictor allow agents to accurately infer the trajectory class from local observations. Experiments show that after training, the agents' prediction accuracy for trajectory classes can stabilize at a high level (Figures 7, 8), enabling the policy to make class-related decisions based on the class representation.
4. Ablation studies and parameter analysis investigate the impact of main modules and hyperparameters (Figures 11, 12, 18), enhancing the interpretability and reproducibility of the method.

**Weaknesses:**

1. The paper does not theoretically analyze the superiority of TRAMA. The authors only demonstrate the advantages of TRAMA over other methods from experimental results, lacking rigorous theoretical derivation and complexity analysis. For example, does the introduction of trajectory clustering and class prediction modules significantly increase the computational overhead? No quantitative analysis is provided.
2. The experimental evaluation is not comprehensive enough. The paper only constructs and tests 4 multi-goal tasks in the StarCraft II environment. However, there are many ways to compose multi-goal tasks. Can these 4 tasks represent the main scenarios in the real world? Moreover, the authors only compare TRAMA with a few representative MARL algorithms. Are there other multi-goal generalization methods that are not included in the comparison? More benchmarks and baselines would make the experimental results more convincing.
3. The number of trajectory classes is predetermined, and the paper does not provide a principled method for setting the classes. It only discusses the impact of different class numbers on performance in the ablation study. So how to adaptively determine the optimal number of classes based on task characteristics? What are the criteria for class division? These questions require further exploration.
4. This paper only focuses on tasks with discrete action spaces. Is it equally applicable to continuous action spaces? Additionally, in other types of multi-agent tasks, such as mixed cooperative-competitive tasks, is trajectory clustering still effective? These issues need to be analyzed and validated in future work.

**Questions:**

1. The paper does not theoretically analyze the superiority of TRAMA. The authors only demonstrate the advantages of TRAMA over other methods from experimental results, lacking rigorous theoretical derivation and complexity analysis. For example, does the introduction of trajectory clustering and class prediction modules significantly increase the computational overhead? No quantitative analysis is provided.
2. The experimental evaluation is not comprehensive enough. The paper only constructs and tests 4 multi-goal tasks in the StarCraft II environment. However, there are many ways to compose multi-goal tasks. Can these 4 tasks represent the main scenarios in the real world? Moreover, the authors only compare TRAMA with a few representative MARL algorithms. Are there other multi-goal generalization methods that are not included in the comparison? More benchmarks and baselines would make the experimental results more convincing.
3. The number of trajectory classes is predetermined, and the paper does not provide a principled method for setting the classes. It only discusses the impact of different class numbers on performance in the ablation study. So how to adaptively determine the optimal number of classes based on task characteristics? What are the criteria for class division? These questions require further exploration.
4. This paper only focuses on tasks with discrete action spaces. Is it equally applicable to continuous action spaces? Additionally, in other types of multi-agent tasks, such as mixed cooperative-competitive tasks, is trajectory clustering still effective? These issues need to be analyzed and validated in future work.

---

> ### Author Response · Authors · 2024-11-24
> **Response #1**
>
> **Q1. Regarding theoretical support and computational cost.** The paper does not theoretically analyze the superiority of TRAMA. The authors only demonstrate the advantages of TRAMA over other methods from experimental results, lacking rigorous theoretical derivation and complexity analysis. For example, does the introduction of trajectory clustering and class prediction modules significantly increase the computational overhead? No quantitative analysis is provided.
>
> **A1.** The reasoning behind the superiority of the conditional policy of TRAMA is based on well-adopted approaches, such as task division methods in Section 4.1 or multi-task reinforcement learning. Task division methods often identify subtasks, roles, or skills first and then determine actions based on these additional conditions along with partial observations. In multi-task reinforcement learning, task ID provided by the environment is often utilized as an additional condition to learn task-dependent policy.
>
> Similar to the above approaches, we presume that this additional condition could help address multi-agent multi-task problems, which are the primary focus of our paper. However, in our multi-task settings, task ID is not provided by the environment. Thus, TRAMA clusters trajectories into certain classes and then each agent predict task types based on its partial observation. Finally, they utilize this additional condition for decision-making to encourage task-dependent policies.
>
> **(computational cost analysis)** Although the overall training time is presented in Appendix B.4 in the original manuscript, we additionally present how each component takes additional computational load in **Appendix B.4** in the revised manuscript, considering the reviewer's comment. The results illustrate that introducing trajectory class and class prediction module does not increase much of additional computation compared to other baseline methods.
>
> **Q2. Regarding further evaluation.** The experimental evaluation is not comprehensive enough. The paper only constructs and tests 4 multi-goal tasks in the StarCraft II environment. However, there are many ways to compose multi-goal tasks. Can these 4 tasks represent the main scenarios in the real world? Moreover, the authors only compare TRAMA with a few representative MARL algorithms. Are there other multi-goal generalization methods that are not included in the comparison? More benchmarks and baselines would make the experimental results more convincing.
>
> **A2-1.** Thanks for the comment. Although four examples of multi-task problems presented in the manuscript cannot represent all possible multi-task settings in the real world, they can evaluate how TRAMA captures some task similarities.
>
> One of key contributions of TRAMA is that it can automatically identifies some similarities and differences among trajectories experienced during training and cluster them into several classes, which are unknown at the beginning. Even though the assumed number of classes is larger than the actual meaningful number of tasks, TRAMA can identify the meaningful number of clusters, as illustrated by **Figure 13** in the manuscript. The reason we introduce customized multi-task problems in Table 1 in the manuscript is to see how effectively TRAMA identifies meaningful clustering.
>
> In addition, we want to clarify that SMACv2 can also be viewed as multi-task settings, according to the definition in Section 2.2 and in the sense that it contains randomly drawn unit combinations and highly different initial unit positions. Although its multiple tasks in original SMACv2 tasks are not clearly defined, TRAMA effectively identifies some similarities and differences between tasks, as illustrated by distinct branches in **Figure 24** in the revised manuscript.
>
> (baseline methods) Although we already include representative and strong baselines, considering the reviewer's comment, we additionally include a baseline called Updet [1] suggested by reviewer YpdJ and another recent work called RiskQ [2]. We present an additional performance comparison with these baselines in **Appendix D.2** of the revised manuscript. **Figures 18-21 in Appendix D.2** illustrate TRAMA's superior performance.
>
> To further evaluate TRAMA, we also conduct out-of-distribution tests to see the effectiveness of TRAMA in unseen tasks. The corresponding results are presented in **Appendix D.6** of the revised manuscript, illustrating TRAMA's superior performance compared to other baseline methods.
>
> [1] Hu, Siyi, et al. "Updet: Universal multi-agent reinforcement learning via policy decoupling with transformers." arXiv preprint arXiv:2101.08001 (2021).
>
> [2] Shen, Siqi, et al. "RiskQ: risk-sensitive multi-agent reinforcement learning value factorization." Advances in Neural Information Processing Systems 36 (2023): 34791-34825.

---

> ### Author Response · Authors · 2024-11-24
> **Response #2**
>
> **Q3. Regarding possible improvement on clustering method.** The number of trajectory classes is predetermined, and the paper does not provide a principled method for setting the classes. It only discusses the impact of different class numbers on performance in the ablation study. So how to adaptively determine the optimal number of classes based on task characteristics? What are the criteria for class division? These questions require further exploration.
>
> **A3.** Thanks for the comment. In **Section 5.3**, we present some guidance to choose $n_{cl}$. If there is information of distinct unit combinations in multi-task settings, it is recommended to select sufficient $n_{cl}$ considering the diversity of unit combinations. Even though $n_{cl}$ exceeds the number of true tasks in multi-task settings, TRAMA captures similarity and additional cluster may share the task types as illustrated by **Figure 13** of the manuscript. In this way, TRAMA still generates coherent conditions.
>
> In addition, we implemented an adaptive clustering module, considering the reviewer's comment. This is one possible implementation, and it selects the best class number, among various $n_{cl}$ candidates based on their Silhouette score. **Appendix D.5** presents the results, demonstrating that the proposed adaptive method effectively identifies the optimal $n_{cl}$, achieving performance comparable to the case with a fixed $n_{cl}$ determined through a parametric search.
>
> **Q4. Regarding possible future work.** This paper only focuses on tasks with discrete action spaces. Is it equally applicable to continuous action spaces? Additionally, in other types of multi-agent tasks, such as mixed cooperative-competitive tasks, is trajectory clustering still effective? These issues need to be analyzed and validated in future work.
>
> Thank you for the comment. Since agents are additionally conditioned on the trajectory-class representation along with the original observation input, it is possible to extend the proposed conditional policy to tasks with continuous action spaces. As this conditional policy promotes coordinated behaviors based on trajectory types, TRAMA could benefit tasks requiring diverse coordination. This presents an interesting direction for future work.

---

### Official Review · Reviewer_9FqX · 2024-11-04

**Soundness:** 2
**Presentation:** 3
**Contribution:** 2
**Rating:** 6
**Confidence:** 4

**Summary:**

The paper introduces a new multi-agent reinforcement learning framework called TRAMA which aims to address the challenge of multi-goal tasks. The multi-goal tasks are especially challenging since they require agents to adapt to goals without relying on task-specific policies. TRAMA incorporates the vector quantize variational autoencoder (VQ-VAE) approach to create a discrete space for trajectory embeddings. With clustering, the embeddings provide information on which trajectory class agents are experiencing and then can be used to improve the decision making process. Experiments are conducted over a suite of SMACv2 tasks and the proposed method achieves superior performance.

**Strengths:**

The paper is well written and easy to follow. The approach to incorporate VQ-VAE for identifying goals is a novel branch of study in MARL. The authors conducted extensive experiments demonstrating the capability of the model. Notably the paper provides insightful visualizations, such as the VQ codebook visualization, which makes the result more convincing. Overall the proposed method achieves a significant improvement in performance in many different settings in SMACv2.

**Weaknesses:**

1. One major concern I have is regarding the novel contribution of TRAMA as compared to the LAGMA framework. Since LAGMA proposed adapting the VQ-VAE framework to multi-agent RL, the paper may benefit from a more detailed discussion on the unique contributions brought by TRAMA.
2. The motivation behind applying k-means clustering to the VQ-VAE embeddings requires further clarification. Since the embeddings are already quantized, it would be helpful to further explain why the k-means clustering is still necessary
3. Following the previous point, the authors may consider providing additional insights into the stability of the k-means clustering process. For me while the result in figure 13 is indeed encouraging, it is in the meantime counterintuitive to see why VQ-VAE embeddings consistently form clear clusters. Additional explanation would strengthen the arguments.

**Questions:**

1. Since the clustering process is trained on a dataset, would the model be able to detect or adapt to out-of-distribution trajectories which better simulate real-life applications?
2. Given the unsupervised nature of VQ-VAE learning and trajectory clustering, how can the model ensure that the resulting clusters are distinct and meaningful with respect to specific tasks?

---

> ### Author Response · Authors · 2024-11-24
> **Response #1**
>
> Thank you for your valuable comments. Please review the global response before proceeding to the detailed responses below.
>
> **W1. Regarding contribution of TRAMA compared to LAGMA.** One major concern I have is regarding the novel contribution of TRAMA as compared to the LAGMA framework. Since LAGMA proposed adapting the VQ-VAE framework to multi-agent RL, the paper may benefit from a more detailed discussion on the unique contributions brought by TRAMA.
>
> **A1.** Thanks for the comment. The major difference is that TRAMA is designed to solve multi-agent multi-task problems, which accompany different unit combinations and highly perturbed initial positions within training. To this end, TRAMA modifies VQ-VAE learning to distribute quantized vectors more evenly across embedding space. In addition, TRAMA introduces a conditional policy based on the trajectory-class type based on agent-wise prediction. To train this predictor model, we adopt clustering methods during centralized training.
>
> In addition to differences in the conditional policy and clustering module, we have updated **Figure 4 and added Appendix D.4** in the revised manuscript, in response to the reviewer's suggestion, to emphasize the significance of TRAMA in adopting VQ-VAE for constructing a quantized latent space.
>
> In addition, the summary and key differences between TRAMA and LAGMA are listed as follows.
>
> (1) Modified coverage loss: TRAMA improves the coverage loss to distribute quantized vectors uniformly across the embedding space. This is critical in multi-task problems, as **Figure 4 and add Appendix D.4** in the revised manuscript illustrate.
>
> (2) Trajectory class identification: In TRAMA, there are two types of trajectory class identification. a) trajectory clustering to identify trajectories that share the commonalities and generate trajectory class labels. These trajectory class labels are utilized for training an 2) agent-wise trajectory class predictor, identifying which trajectory class agents are experiencing based on their partial observation. This predicted information is used as prior knowledge or additional conditions in trajectory-class-dependent policy.
>
> (3) Trajectory-class-dependent policy: While LAGMA resorts to latent-goal guided incentive based on visitation on quantized vectors, TRAMA utilizes predicted trajectory-class information for agent-wise decision-making and learns policy versatile to multi-task settings.
>
> **W2. Regarding the necessity of clustering.** The motivation behind applying k-means clustering to the VQ-VAE embeddings requires further clarification. Since the embeddings are already quantized, it would be helpful to explain further why the k-means clustering is still necessary.
>
> **A2.** Thanks for the comment. Additional clustering is required to generate trajectory-class labels since the environment does not provide class labels, even though agents operate in multi-task settings. Without supervision by labels from the environment, TRAMA automatically identifies commonalities among trajectories and conducts clustering based on this identification. By constructing trajectory embedding with quantized vectors, TRAMA can capture key features of trajectories classified as the same class while neglecting some differences in the original state space. In this way, TRAMA can yield more distinct clustering with quantized vectors.
>
> **W3. Regarding clustering method (cont').** Following the previous point, the authors may consider providing additional insights into the stability of the k-means clustering process. For me while the result in figure 13 is indeed encouraging, it is in the meantime counterintuitive to see why VQ-VAE embeddings consistently form clear clusters. Additional explanation would strengthen the arguments.
>
> **A3.** As explained in A2, trajectory embedding constructed by VQ-VAE embeddings can capture key features among trajectories while neglecting some differences in the original state space. Thus, TRAMA can generate more coherent clustering results. Hence, having VQ-VAE embeddings well distributed across the overall embedding space $\chi = \\{x \in \mathbb{R}^d : x = f_{\phi}^e(s), s \in \mathcal{D}\\}$ is important to generate coherent and clear clusters. That is why we further modify the loss function of VQ-VAE training in Eq. 6 of the original manuscript. Again, **Figure 4 and added Appendix D.4** present how clustering is influenced by the distribution of VQ-VAE embedding vectors.

---

> ### Author Response · Authors · 2024-11-24
> **Response #2**
>
> **Q1. Regarding further generalizability.** Since the clustering process is trained on a dataset, would the model be able to detect or adapt to out-of-distribution trajectories which better simulate real-life applications?
>
> **A4.** Thanks for the really valuable and interesting comment. In response to the reviewer's suggestion, we conduct additional experiments and corresponding qualitative analysis on out-of-distribution tasks in **Appendix D.6** of the revised manuscript.
>
> The following results summarize the problem settings and corresponding results. For this evaluation, we first trained TRAMA and baseline models in a given multi-task problem (SurComb4) and tested them in OOD tasks presented in the following Table.
>
> # Task Configuration of OOD Tests
> | Name       | Initial Position Type    | Unit Combinations           |
> |------------|--------------------------|-----------------------------|
> | ID (SurComb4) | Surrounded              | {1c2s2z, 3s2z, 2c3z, 2c3s} |
> | OOD (#1)   | Surrounded               | {1c4s, 1c3s1z, 2s3z}        |
> | OOD (#2)   | Surrounded               | {5s, 5z, 5c}                |
> | OOD (#3)   | Surrounded and Reflected | {1c2s2z, 3s2z, 2c3z, 2c3s} |
> | OOD (#4)   | Surrounded and Reflected | {1c4s, 1c3s1z, 2s3z}        |
> | OOD (#5)   | Surrounded and Reflected | {5s, 5z, 5c}                |
>
> # Return of OOD Tests
> |            | TRAMA     | QMIX   | QPLEX  | EMC    |
> |------------|-----------|--------|--------|--------|
> | ID (SurComb4) | **3.063**$^*$ ± 0.489 | 2.201 ± 0.398 | 2.323 ± 0.119 | 2.348 ± 0.406 |
> | OOD (#1)   | **2.706**$^*$ ± 0.461 | 2.077 ± 0.422 | 2.248 ± 0.471 | 1.661 ± 0.220 |
> | OOD (#2)   | **1.259**$^*$ ± 0.148 | 0.939 ± 0.463 | 1.049 ± 0.240 | 0.774 ± 0.326 |
> | OOD (#3)   | **2.307**$^*$ ± 0.488 | 1.899 ± 0.299 | 2.025 ± 0.221 | 2.153 ± 0.557 |
> | OOD (#4)   | **2.317**$^*$ ± 0.280 | 1.484 ± 0.219 | 1.993 ± 0.199 | 1.457 ± 0.127 |
> | OOD (#5)   | **0.921**$^*$ ± 0.679 | 0.532 ± 0.200 | 0.792 ± 0.282 | 0.605 ± 0.295 |
>
> # Win-rate of OOD Tests
> |            | TRAMA     | QMIX   | QPLEX  | EMC    |
> |------------|-----------|--------|--------|--------|
> | ID (SurComb4) | **0.707**$^*$ ± 0.025 | 0.540 ± 0.059 | 0.667 ± 0.034 | 0.613 ± 0.025 |
> | OOD (#1)   | **0.625**$^*$ ± 0.057 | 0.475 ± 0.062 | 0.525 ± 0.073 | 0.430 ± 0.057 |
> | OOD (#2)   | **0.280**$^*$ ± 0.020 | 0.270 ± 0.057 | 0.265 ± 0.017 | 0.275 ± 0.057 |
> | OOD (#3)   | **0.620**$^*$ ± 0.111 | 0.465 ± 0.050 | 0.545 ± 0.050 | 0.525 ± 0.084 |
> | OOD (#4)   | **0.545**$^*$ ± 0.052 | 0.355 ± 0.071 | 0.475 ± 0.059 | 0.365 ± 0.033 |
> | OOD (#5)   | **0.220**$^*$ ± 0.141 | 0.160 ± 0.037 | 0.215 ± 0.050 | 0.205 ± 0.062 |
>
> As the results in above Tables illustrate, TRAMA shows strong performance in OOD tasks containing some similarity (OOD\#1 and OOD\#4) with in-distribution tasks. We speculate the reason as TRAMA gains some benefits from identifying similar tasks from experience during training and encouraging similar policies through trajectory-class representation. In this way, TRAMA can adapt to OOD tasks. On the other hand, for tasks with no apparent task similarity (OOD\#2 and OOD\#4), the performance gain decreases.
>
> In addition, when there is no clear task similarity (highly out-of-distributed tasks), the prediction confidence of agents-wise prediction decreases. In this case, agents can detect OOD tasks or trajectories by setting some threshold value for confidence. Please see details in **Figures 31(c)** of the revised manuscript.
>
> **Q2. Regarding clustering.** Given the unsupervised nature of VQ-VAE learning and trajectory clustering, how can the model ensure that the resulting clusters are distinct and meaningful with respect to specific tasks?
>
> **A5.** The key component to construct meaningful clustering is distributing quantized vectors evenly across the embedding space. If the embedding space $\chi = \\{ x \in \mathbb{R}^d : x = f_{\phi}^e(s), s \in \mathcal{D} \\}$ contains distinct branches, i.e., multiple tasks as illustrated by Figure 4 of the revised manuscript (or Figure 3 in the original manuscript), VQ-VAE vectors considering trajectory-class signifies the difference.
> This is because both embedding result $x$ and the nearest quantized embedding vector $e = [x]_q$ attract each other by
>
> $\mathcal{L}_{VQ}^{tot}(\phi, \mathbf{e})$ of Eq. 5. Through this mechanism, when there exist distinct trajectory difference, TRAMA can effectively capture this distinction. However, without considering trajectory classes in $\mathcal{J}$, it is hard to ensure that the model generates distinct and meaningful clusters as illustrated by Figure 4 in the revised manuscript and Appendix D.4.

---

> > ### Comment · Reviewer_9FqX · 2024-11-27
> > **Thank you for the rebuttal**
> >
> > I would like to thank the authors for the great efforts in the detailed explanation and additional experiments. Most of my concerns have been addressed and I will keep my current score.

---

### Official Review · Reviewer_8EYy · 2024-11-16

**Soundness:** 2
**Presentation:** 1
**Contribution:** 2
**Rating:** 5
**Confidence:** 3

**Summary:**

This paper focuses on utilizing multi-agent reinforcement learning (RL) for solving the multi-goal task. An approach of TRajectory-class-Aware Multi-Agent reinforcement learning (TRAMA) is proposed by generating the trajectory embeddings, making trajectory clustering, learning trajectory-class-aware policy with trajectory-class representation. Finally, experiments were conducted on SMACv2 domain, including four modified tasks (multi-goal tasks) and two conventional SMACv2 tasks, and compared TRAMA with QMIX, RODE, LDSA,  MASER, QPLEX, EMC, and LAGMA.

**Strengths:**

- The multi-agent RL has the potential to solve more complex tasks. This paper aims to use multi-agent RL to solve the multi-goal task. The main contribution of this paper is by proposing a framework with doing clustering and prediction on the latent space and learning the policies based on the generated trajectory-class representation. Although the main idea is straightforward, there is some novelty in the algorithm design.

**Weaknesses:**

- The presentation should be improved. What's the formal definition of the multi-goal task? Why multi-agent RL can solve this better than the single-agent RL? What's the relationship between the multi-goal and trajectories? In Figure 2, what's the difference between "$s_t$" and "$o_t^i$"? what are the goals? How different are they? How to formulate them into the objective? How agents interact with each other and the environment, and learn various goals? Is the whole training process end-to-end? or are there some pre-training process? I do believe there needs some clarifications on the problem background, assumptions, and details of the methodology, currently it's not clear to me.
- Empirical evaluation: what's the difference between the modified tasks and conventional tasks on SMACv2? Are they all multi-goal settings? What are these goals? In the results shown in Figure 6, it looks the improvement of TRAMA over others is marginal. For Figures 7 and 8, it's better to report the numbers instead of showing the curves. In conventional tasks on SMACv2, it looks TRAMA barely didn't have the advantage over others. If this is the case, how could TRAMA work better on multi-goal tasks than other methods? It feels like the methodology is quite complex, but the benefit is small. Overall, the current conclusion is still vague to me.

**Questions:**

- see my comments in the weaknesses.
- Did the replay buffer store the trajectories only, instead of transitions?
- How many agents were used? what are the number of trajectory classes and centroid? How did you select them?
- How did you collect the trajectories? What's size of the replay buffer?

---

> ### Author Response · Authors · 2024-11-24
> **Response #1**
>
> Thank you for your valuable comments. Please review the global response before proceeding to the detailed responses below.
>
>  **Weakness 1. Regarding presentation and problem settings.**
>
>  **A1.** Thank you for the valuable comments. To follow the reviewer's comment, we present **the formal definition of multi-task settings in Section 2.2 of the revised manuscript**. In addition, we revised the ambiguity presented in the original Figure 2, to present "$s_t$" and "$o_t^i$" separately. As presented in the preliminary section (Sec. 2.1), $s$ represents a global state while $o^i$ represents a partial observation of agent $i$.
>
> (why MARL is better?) For clarification, this paper does not claim that the multi-agent RL approach is better than the single-agent approach. Instead, we are addressing the problem in different settings, i.e., DecPOMDP. We would like to clarify this point first. If anything in the manuscript misled you, please let us know which part caused the confusion. In addition, in single-agent goal-conditioned RL or multi-task RL, a goal or task ID is often clearly given, and an agent accesses to full state information. On the other hand, in MARL, agents only access to partial information and experience various tasks without knowledge of task types in multi-task settings we solve in this paper.
>
> (Training and interaction with the environment) Yes. TRAMA is trained in end-to-end, without any pre-training. Initially, no conditional information is provided by the environment. Through centralized training, TRAMA identifies trajectory classes based on trajectory embedding. Then, agents predict which trajectory class they are experiencing based on their partial observation and then use it as additional conditional information. In this way, agents can learn a generalizable policy to various tasks.
>
>  **Weakness 2-1. Regarding empirical evaluation.**
>
> **A2-1.** As presented in Section 5, in modified tasks, unit combinations are randomly selected from the designated sets presented in Table 1 of the revised manuscript. On the other hand, in the conventional tasks in SMACv2, units are randomly drawn from possible units. Appendix B in the revised manuscript details multi-task problems and SMACv2. In addition, as explained in Section 5, another difference between them is that the customized multi-task problems adopt sparse reward settings presented in Table 2 in Appendix B.1 of the original or revised manuscript. We defer the details to the appendix due to the page limit.
>
> Yes. They are all viewed as multi-task settings according to the definition of multi--tasks in Section 2.2, as both tasks provide different unit combinations and highly different initial positions compared to conventional SMAC tasks.
>
>  TRAMA shows the consistently best performance in various settings in **Figure 7 and Figure 9** of the revised manuscript. Given these coherent results and challenges posed by multi-task problems, we deem the performance improvement of TRAMA to be both promising and significant, despite requiring additional modules.
>
> Thanks for the comment. We consider the variation in prediction accuracy over training time as an important factor, and the graph provides a more intuitive representation in this regard. However, in response to the reviewer's comment, we present compact graph results in **Figure 10** and detailed results in Tables in **Appendix D.7** of the revised manuscript.

---

> ### Author Response · Authors · 2024-11-24
> **Response #2**
>
> **Weakness 2-2. Regarding empirical evaluation (cont').**
>
>  **A2-2.** To clarify, TRAMA shows better performance compared to other baselines in the conventional SMACv2 task, as presented in Figure 7 of the revised manuscript. The review may have mentioned the results from the conventional SMAC task presented in Figure 8, which is a single-task problem. Although TRAMA is designed to solve multi-task problems, it also shows comparable or better performance in single-task settings like SMAC.
>
>  In addition, please note that the performance of conventional SMAC tasks in various maps is quite saturated, yielding marginal differences in performance among various methods. Thus, researchers introduced more challenging problems, such as SMACv2.
>
> To evaluate TRAMA, we additionally present the experiment results in **Appendix D.6**, illustrating the generalizability of TRAMA in out-of-distribution tasks. The results demonstrate the strong performance of TRAMA in OOD tasks, by identifying similar tasks among in-distribution tasks and conditioning on these predictions during decision-making, thereby encouraging a joint policy that benefits OOD tasks.
>
> # Task Configuration of OOD Tests
> | Name       | Initial Position Type    | Unit Combinations           |
> |------------|--------------------------|-----------------------------|
> | ID (SurComb4) | Surrounded              | {1c2s2z, 3s2z, 2c3z, 2c3s} |
> | OOD (#1)   | Surrounded               | {1c4s, 1c3s1z, 2s3z}        |
> | OOD (#2)   | Surrounded               | {5s, 5z, 5c}                |
> | OOD (#3)   | Surrounded and Reflected | {1c2s2z, 3s2z, 2c3z, 2c3s} |
> | OOD (#4)   | Surrounded and Reflected | {1c4s, 1c3s1z, 2s3z}        |
> | OOD (#5)   | Surrounded and Reflected | {5s, 5z, 5c}                |
>
> # Return of OOD Tests
>
> |            | TRAMA     | QMIX   | QPLEX  | EMC    |
> |------------|-----------|--------|--------|--------|
> | ID (SurComb4) | **3.063**$^*$ ± 0.489 | 2.201 ± 0.398 | 2.323 ± 0.119 | 2.348 ± 0.406 |
> | OOD (#1)   | **2.706**$^*$ ± 0.461 | 2.077 ± 0.422 | 2.248 ± 0.471 | 1.661 ± 0.220 |
> | OOD (#2)   | **1.259**$^*$ ± 0.148 | 0.939 ± 0.463 | 1.049 ± 0.240 | 0.774 ± 0.326 |
> | OOD (#3)   | **2.307**$^*$ ± 0.488 | 1.899 ± 0.299 | 2.025 ± 0.221 | 2.153 ± 0.557 |
> | OOD (#4)   | **2.317**$^*$ ± 0.280 | 1.484 ± 0.219 | 1.993 ± 0.199 | 1.457 ± 0.127 |
> | OOD (#5)   | **0.921**$^*$ ± 0.679 | 0.532 ± 0.200 | 0.792 ± 0.282 | 0.605 ± 0.295 |
>
> **Q1. see my comments in the weaknesses.**
>
> **A3.** Please see the answer A1 - A2 above.
>
> **Q2. Regarding the construction of the replay buffer.** Did the replay buffer store the trajectories only, instead of transitions?
>
> **A4.** In MARL settings, the episodic trajectory information, including transitions at each timestep $\<\boldsymbol{o}_0, \boldsymbol{a}_0, r_0, \boldsymbol{o}_1, \dots, \boldsymbol{o}_T\>$ and global states $\<s_0, s_1, s_2, \dots, s_T\>$
> , is stored in the replay buffer. Please note that global state $s_t$ can only be used during centralized training, while $o_t^i$ is used during decentralized execution. These are the standard CTDE settings presented in Appendix A.3 of the revised manuscript or Section 2.2 of the original manuscript.

---

> ### Author Response · Authors · 2024-11-24
> **Response #3**
>
> **Q3. Regarding experiment setting and selection of the number of trajectory classes.**  How many agents were used? what are the number of trajectory classes and centroid? How did you select them?
>
> **A5.** In MARL settings, the number of agents is often described by "task name." For example, p5\_vs\_5 means 5 ally agents (agents to be trained) compete with 5 enemy agents (not objects of training). Similarly, 3s2z means that (5=3+2) ally agents compete with (5=3+2) enemy agents.
>
> For convenience, we added the number of training agents in **Table 3 of Appendix B.1** of the revised manuscript and additional explanation accordingly in Appendix B.1. Please refer to the modified manuscript.
>
> (Regarding clustering) The number of trajectory classes and centroids used for each task are presented throughout the paper with the notation $n_{cl}$. In addition, the task-dependent $n_{cl}$ and other hyperparameters are summarized in Table 4 of the revised manuscript.
>
> Thanks for the comment. Selecting $n_{cl}$ is not so obvious, but we present some guidance for it in **Section 5.3** in the manuscript. Furthermore, regardless of the selection of $n_{cl}$, TRAMA could capture distinct and meaningful trajectory classes well, as illustrated by **Section 5.5 and Figure 13**, where meaningful four branches of trajectory classes were identified with initial $n_{cl}=8$.
>
> For centroid initialization, centroids are initially selected randomly from the data points. Then next centroid is selected probabilistically, where the probability of selecting a point is proportional to the square of its distance from the nearest existing centroid. At first, we iterate Kmeans with 10 different initial centroids. However, as discussed in Section 3.2, generating coherent labels is also important for training trajectory-class predictor. Thus, once we get the previous centroid value, we use this prior value as an initial guess for the centroid and run K-means just once with it.
>
> In addition, we present a possible adaptive clustering method in **Appendix D.5**, which updates the number of cluster $n_{cl}$ based on Silhouette scores of $n_{cl}$ candidates.
>
> **Q4. Regarding the construction of the replay buffer (cont')**. How did you collect the trajectories? What's size of the replay buffer?
>
>  **A7.** As presented in Line \#7 of Algorithm 2 in Appendix C (original manuscript), agents interact with the environment with a given policy. Once we get the episodic trajectory, each trajectory is stored in the replay buffer in FIFO style. We adopt the same replay buffer size as in other baseline methods [1,2,3], size of 5000.
>
> [1] Rashid, Tabish, et al. "Monotonic value function factorisation for deep multi-agent reinforcement learning." Journal of Machine Learning Research 21.178 (2020): 1-51.
>
> [2] Wang, Jianhao, et al. "Qplex: Duplex dueling multi-agent q-learning." arXiv preprint arXiv:2008.01062 (2020).
>
> [3] Zheng, Lulu, et al. "Episodic multi-agent reinforcement learning with curiosity-driven exploration." Advances in Neural Information Processing Systems 34 (2021): 3757-3769.

---

### Meta-Review · Area_Chair_r6E6 · 2024-12-22

**Metareview:**

The paper proposes a new multi-agent RL approach, TRAMA that enables a set of agents to adapt to a task without being given the task ID. The key idea is to encode the agents' observation history into a latent space that enables the agents to recognize the task implicitly and decide how they each of them should behave accordingly. In the paper's TRAMA shows superior performance in several StarCraft II scenarios.

Strengths:

- Novelty in the context of MARL.
- Clarity (in the post-discussion version of the submission).

Weaknesses:

- Comprehensiveness of the evaluation.
- The motivation for clustering (the fact that the agents aren't told what task they are solving).

Overall, the reviewers consider the method novel despite combining well-known techniques like VQ-VAE-based trajectory embeddings and generally consider it a valuable contribution to MARL research. The metareviewer won't override their collective opinion in this case. However, to someone from a closely related research community but outside the MARL community itself, the problem setting the paper focuses on can look highly artificial. Why aren't the agents told what task they are solving? In what real settings is this a reasonable assumption? The paper makes no attempt to explain this. This could be addressed by describing in detail the tasks that this work uses for the experiments, but the paper merely refers to the benchmark, mentions the task names, and says that the tasks come from the StarCraft II environment, without clarifying what the tasks actually are, even in Appendix B. Given that ICLR isn't a specialized MARL conference, the metareviewer strongly encourages the authors to add a much more explicit motivation for problem setting.

**Additional Comments On Reviewer Discussion:**

The discussion has definitely helped improve the paper's clarity: based on the original submission's description, the problem setting was non-obvious to the reviewers. In the process, the need for TRAMA's clustering step was clarified as well (it is needed because the agents don't get the ground-truth labels of the tasks they are solving). The discussion also highlighted the merits of the existing evaluation, even though the reviewers' original criticism that it could be more extensive still stands, in the metareviewer's opinion.

---

### Decision · Program_Chairs · 2025-01-22

Accept (Poster)